# Low phosphate activates STOP1-ALMT1 to rapidly inhibit root cell elongation

Coline Balzergue[1,2,3,*,†], Thibault Dartevelle[1,2,3,*], Christian Godon[1,2,3,*], Edith Laugier[1,2,3,*],
Claudia Meisrimler[1,2,3,*,†], Jean-Marie Teulon[4,5,6], Audrey Creff[1,2,3,†], Marie Bissler[1,2,3], Corinne Brouchoud[1,2,3],
Agnès Hagège[7,†], Jens Müller[8], Serge Chiarenza[1,2,3], Hélène Javot[1,2,3], Noëlle Becuwe-Linka[1,2,3],
Pascale David[1,2,3], Benjamin Péret[1,2,3,†], Etienne Delannoy[1,2,3,†], Marie-Christine Thibaud[1,2,3], Jean Armengaud[9],
Steffen Abel[8], Jean-Luc Pellequer[4,5,6], Laurent Nussaume[1,2,3] & Thierry Desnos[1,2,3]

Environmental cues profoundly modulate cell proliferation and cell elongation to inform and direct plant growth and development. External phosphate (Pi) limitation inhibits primary root growth in many plant species. However, the underlying Pi sensory mechanisms are unknown. Here we genetically uncouple two Pi sensing pathways in the root apex of *Arabidopsis thaliana*. First, the rapid inhibition of cell elongation in the transition zone is controlled by transcription factor STOP1, by its direct target, *ALMT1*, encoding a malate channel, and by ferroxidase LPR1, which together mediate Fe and peroxidase-dependent cell wall stiffening. Second, during the subsequent slow inhibition of cell proliferation in the apical meristem, which is mediated by LPR1-dependent, but largely STOP1–ALMT1-independent, Fe and callose accumulate in the stem cell niche, leading to meristem reduction. Our work uncovers STOP1 and ALMT1 as a signalling pathway of low Pi availability and exuded malate as an unexpected apoplastic inhibitor of root cell wall expansion.

[1] Laboratoire de Biologie du Développement des Plantes, Institut de Biosciences et Biotechnology Aix-Marseille, Commissariat à l'Energie Atomique et aux énergies alternatives, Saint-Paul-Lez-Durance 13108, France. [2] Centre National de la Recherche Scientifique, UMR 7265 Biol. Végét. & Microbiol. Environ., Saint-Paul-Lez-Durance, France. [3] Aix-Marseille Université, UMR 7265, Marseille, France. [4] CNRS, IBS, Grenoble F-38044, France. [5] CEA, IBS, Grenoble F-38044, France. [6] Université Grenoble Alpes, IBS, Grenoble F-38044, France. [7] Commissariat à l'Energie Atomique et aux énergies alternatives, Service de Biologie et de Toxicologie Nucléaire, Laboratoire d'Etude des Protéines Cibles, 30200 Bagnols sur Cèze, France. [8] Department of Molecular Signal Processing, Leibniz Institute of Plant Biochemistry, Halle (Saale) 06120, Germany. [9] CEA, DRF, JOLIOT/DMTS/SPI/Li2D, Laboratory 'Innovative Technologies for Detection and Diagnostics', Bagnols-sur-Cèze F-30200, France. * These authors contributed equally to this work. † Present addresses: Laboratoire de Recherche en Sciences Végétales, Université de Toulouse, CNRS, UPS, 24 chemin de Borde Rouge, Auzeville, BP42617, 31326 Castanet Tolosan, France (C.B.); Plant-Microbe Interactions, Department of Biology, Faculty of Science, Utrecht University, Padualaan 8, Hugo R. Kruytgebouw, 3584 CH Utrecht, The Netherlands (C.M.); Laboratoire Reproduction et Développement des Plantes, ENS de Lyon, UCB Lyon 1, CNRS, INRA, F-69342 Lyon, France (A.C.); CNRS, Université Claude Bernard Lyon 1, Ens de Lyon, Institut des Sciences Analytiques, UMR 5280, 5 rue de la Doua, F-69100 Villeurbanne, France (A.H.); Centre National de la Recherche Scientifique, Equipe Développement et Plasticité du Système Racinaire, UMR Biochimie et Physiologie Moléculaire des Plantes, Institut de Biologie Intégrative des Plantes, Campus INRA/Montpellier SupAgro, 2 Place Pierre Viala, 34060 Montpellier, France (B.P.); Institute of Plant Sciences Paris Saclay IPS2, CNRS, INRA, Université Paris-Sud, Université Evry, Université Paris-Saclay, Bâtiment 630, 91405 Orsay, France et Institute of Plant Sciences Paris-Saclay IPS2, Paris Diderot, Sorbonne Paris-Cité, Bâtiment 630, 91405 Orsay, France (E.D.). Correspondence and requests for materials should be addressed to T.D. (thierry.desnos@cea.fr).

Phosphate limitation ($-$Pi), prevailing in around 70% of global cultivated land, is a common abiotic stress that profoundly alters root growth with critical consequences for crop yields[1,2]. However, the underlying mechanisms of root growth inhibition by low Pi availability remain largely unknown.

To address this problem at the cellular and molecular level, we chose the primary root of *Arabidopsis thaliana* seedlings as the experimental system because it displays a transparent and simple anatomical structure. Roots longitudinally and indeterminately grow at their apex. Decades of laboratory work have shown that cells proliferate in the root apical meristem (RAM) in concentric layers (epidermis, cortex and endodermis surrounding the stele). These cell types originate in the stem cell niche (SCN), enclosing the quiescent centre (QC), divide as transit-amplifying cells before rapid and extensive cell expansion ensues in the elongation zone (EZ)[3,4] (Supplementary Fig. 1a). A complex set of regulatory networks and developmental genes coordinate cell proliferation and cell elongation required for root extension[4].

Previous studies with *Arabidopsis* seedlings showed that acute $-$Pi stress is locally sensed at the primary root apex in a Fe-dependent manner and inhibits root growth by affecting both cell elongation and cell proliferation[5–12]. The local response is independent of internal root Pi status[13] and is therefore not caused by a metabolic deficiency syndrome.

To date, several mutants specifically altered in the local root response have been isolated[5,6,14–18], but only a few corresponding mutated genes have been identified. Much has been learned from *LOW PHOSPHATE ROOT 1 (LPR1)* and *PHOSPHATE DEFICIENCY RESPONSE 2 (PDR2)*, which genetically interact and are expressed in cell type-specific but overlapping domains of the root apex, comprising the RAM and early EZ[8,9,12]. While *LPR1* encodes a cell wall (CW) targeted multicopper oxidase with ferroxidase activity, *PDR2* codes for AtP5A, the single P5-type ATPase in *Arabidopsis*, which is thought to repress LPR1 biogenesis or activity[8,9]. Under $-$Pi, the genetic *PDR2–LPR1* module mediates accumulation of Fe in the SCN and EZ within 20 h, which triggers generation of reactive oxygen species (ROS) as well as deposition of callose and pectin in CWs of the RAM and EZ[12,19]. Callose deposition in the SCN obstructs cell-to-cell communication as demonstrated by impaired movement of SHORT-ROOT, a key transcription factor of root patterning and cell fate specification[4], which is followed by reduced stem cell maintenance and RAM activity[12].

Here we identify and characterize two key players of a new signalling pathway, which is activated under $-$Pi to rapidly inhibit root growth by targeting cell expansion in the EZ: the transcription factor STOP1 (SENSITIVE TO PROTON TOXICITY1) and the malate efflux channel ALUMINUM-ACTIVATED MALATE TRANSPORTER1 encoded by *ALMT1*, a direct STOP1 target gene. Surprisingly, mutations in either gene spatially and temporally uncouple the rapid inhibition of cell elongation in the EZ ($<$2 h) from the slower inhibition of cell proliferation in the SCN and RAM ($<$4 days), which are both dependent on LPR1 ferroxidase activity. Based on our data, we propose that malate efflux by internal cell files of the root tip facilitates Fe chelation in the apoplast. LPR1-dependent Fe redox cycling stimulates peroxidase-catalysed CW stiffening followed by inhibition of root cell elongation.

## Results

**STOP1 and ALMT1 repress primary root growth under $-$Pi.** To discover new genes important for primary root growth inhibition under $-$Pi, we screened $\sim 4 \times 10^5$ ethylmethane sulfonate-mutagenized WT seedlings (see Methods). We isolated 85 independent mutants with a significantly longer primary root when grown under $-$Pi. Genetic mapping and molecular analysis identified 26 *lpr1* alleles, suggesting that our screen approached saturation for isolating complete or partial loss-of-function mutations conferring a *lpr1*-like root phenotype. We also identified 24 mutant alleles of the transcription factor STOP1 and 14 alleles of the malate efflux channel ALMT1 (see Methods, Fig. 1a–c and Supplementary Fig. 1b), we describe here a representative subset of both complementation groups. All *almt1* and *stop1* mutations are recessive, except for *Stop1[25]* and *Stop1[48]*, which are dominant negative mutations displaying the same phenotype as seedlings homozygous for *stop1* null alleles (Supplementary Fig. 1c,d). The presumably truncated mutant STOP1 proteins dominantly inhibit *ALMT1* expression, possibly by titrating a limiting factor required for *ALMT1* activation.

**ALMT1 promotes primary root growth arrest under $-$Pi.** Previous studies showed that STOP1 promotes root exudation of malate and citrate by stimulating expression of ALMT1 and MATE efflux transporters, respectively[20–24]. Root exudation of organic acids such as malate and citrate is thought to improve plant growth by solubilizing Pi anions complexed with metal cations like iron (Fe) and aluminium (Al) in soil[25,26]. We were therefore intrigued by the observation that *stop1* and *almt1* mutants develop a longer primary root than the WT in $-$Pi conditions. We thus analysed in more detail the role of STOP1 and ALMT1 in Pi-dependent root growth inhibition.

First, chemical complementation by exogenous malate application restored in a dose-dependent manner primary root growth inhibition on $-$Pi medium in *almt1* mutants (Fig. 1d); whereas no reduced root growth was observed in $+$Pi conditions (Supplementary Fig. 2a). This result is consistent with the hypothesis that the ALMT1-dependent malate exudation inhibits WT root growth under $-$Pi.

We then tested whether expression of *ALMT1* and *STOP1* is regulated by Pi availability. One hour after transfer from the $+$Pi to $-$Pi medium the expression level of *ALMT1* increases and reaches a plateau at 24 h (Fig. 1e). However, on both media *ALMT1* expression is abrogated in strong *stop1* mutants (Supplementary Fig. 2b). *MATE* is also upregulated on $-$Pi medium (Fig. 1e) in a STOP1-dependent manner (Supplementary Fig. 2b). However, under $-$Pi the *mate[KO]* mutant does not grow a long primary root, and the main root of the *almt1[61];mate[KO]* double mutant is as long as the root of the *almt1[61]* single mutant (Supplementary Fig. 2c). The pattern of *MATE* expression in the primary root is different from that of *ALMT1*, in particular it is not expressed in the root tip[27], and this may explain why the *mate[KO]* single mutant is not altered in the root growth under $-$Pi. We thus tested whether expressing *MATE* under the control of the *ALMT1* promoter could restore root growth inhibition by $-$Pi of the *almt1[KO];mate[KO]* double mutant. First, we confirmed that in this line[27] the *pALMT1::MATE* construct is functional and complements the $Al^{3+}$ sensitivity of the *almt1[KO];mate[KO]* double mutant (Supplementary Fig. 2d). However, expression of *pALMT1::MATE* in *almt1[KO];mate[KO]* plants did not restore primary root growth inhibition on $-$Pi medium (Supplementary Fig. 2d). Together, these results indicate that citrate exudation by MATE cannot replace malate exudation by ALMT1 to inhibit primary root growth under $-$Pi.

A prominent role of ALMT1 as an effector downstream of STOP1 for root growth inhibition under $-$Pi is supported by the following observations: (i) the root length of *stop1* and *almt1* mutants are similar (Fig. 1c), and (ii) the constitutive expression of *ALMT1* fully rescues both *almt1[32]* (Supplementary Fig. 2e) and the dominant negative *Stop1[48]* mutant phenotypes (Supplementary Fig. 2f).

We quantitatively showed in yeast one-hybrid assays that STOP1 recognizes a small region of the *ALMT1* promoter including the STOP1-binding site found previously by others[28] (Supplementary Fig. 2g). Interestingly, the low Pi-stimulated expression of *ALMT1* is independent of PHR1 and PHL1 (Supplementary Fig. 3a), two master transcription factors controlling many Pi-starvation responses[29].

— **Pi enhances expression of *ALMT1* in the root apex**. To determine the spatial expression pattern of *ALMT1* and *STOP1*, we established several independent transgenic lines carrying the GUS reporter gene placed under the control of their respective promoters (*pALMT1::GUS* and *pSTOP1::GUS*). The *pALMT1::-GUS* lines (WT background) showed *ALMT1* expression in the whole root (Supplementary Fig. 3b). This histological marker

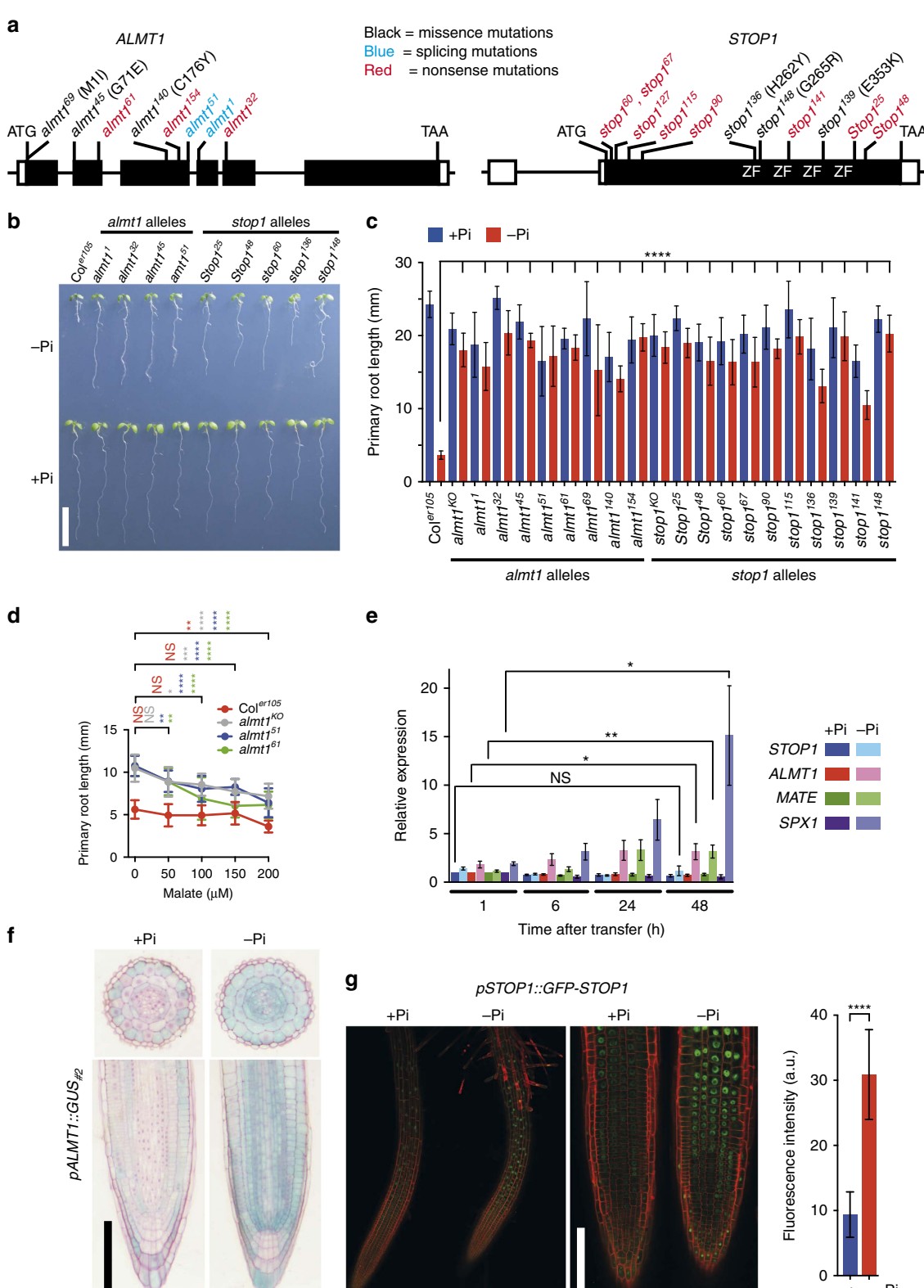

likely reports the genuine expression pattern of endogenous *ALMT1*, confirming our quantitative PCR with reverse transcription (qRT − PCR) analysis (Fig. 1e). As shown in Fig. 1f (and Supplementary Fig. 3b), *pALMT1::GUS* was more highly expressed in − Pi than in + Pi conditions, but was not detectable in the *Stop1*[48] mutant background (Supplementary Fig. 3c).

We previously showed that the root growth arrest does not occur under − Pi at pH >6.3, or when iron is omitted in the medium[8]. We therefore assessed whether this observation is due to the lack of *ALMT1* expression. Our results revealed that *ALMT1* expression was not detectable in roots on + Pi medium at pH 7.1 (Supplementary Fig. 4a), whereas Fe omission did not impair stimulation of *ALMT1* expression by − Pi (Supplementary Fig. 4a). On the other hand, WRKY46, a transcription factor whose expression is enhanced in roots by Fe deficiency[30] and which represses *ALMT1* expression[31], does not seem to modulate the root growth inhibition under − Pi because the *wrky46*[KO] mutant behaved like the WT control (Supplementary Fig. 5). These results support the view that Fe does not play a crucial role in the induction of *ALMT1* by − Pi.

As shown in longitudinal and transverse sections of the root meristem (Fig. 1f), *ALMT1* is expressed during − Pi limitation in the SCN, including the QC, and in all cell files, although at a lower level in the external root cap layer and columella cells. We therefore demonstrated here that, in addition to other cues[32], − Pi stimulates *ALMT1* expression in the root tip.

The *pSTOP1::GUS* marker revealed that *STOP1* is expressed in the entire root tip (Supplementary Fig. 6a); however, in contrast to *ALMT1*, *STOP1* expression is not enhanced by − Pi exposure (Fig. 1e and Supplementary Figs 2b, 3a and 6a). This suggests that STOP1 is post-transcriptionally regulated to stimulate *ALMT1* expression under − Pi. We thus tested whether nuclear localization of STOP1 is regulated. We fused the Green Fluorescent Protein (GFP) to the STOP1 N terminus and the GFP-STOP1 coding region was placed under control of the *pSTOP1* or *pUBQ10* promoter. The translational fusions are functional because they complement the root growth response of the *stop1*[KO] mutant (Supplementary Fig. 6b). Confocal microscopy of root tips revealed nuclear localization of GFP-STOP1 under + Pi (Fig. 1g and Supplementary Fig. 6c), which interestingly intensified on seedling transfer to − Pi medium (Fig. 1g and Supplementary Fig. 6c). These results indicate that STOP1 is post-translationally regulated and suggest that the stimulation of *ALMT1* expression under − Pi is, at least partly, due to increased accumulation of STOP1 in the nucleus. Overall, our results show that the STOP1–ALMT1 module is a component of a new, PHR1-independent regulatory pathway activated by − Pi stress.

**Root growth arrest under − Pi is rapid.** To characterize with higher temporal resolution, the early phase of primary root growth inhibition under − Pi, we performed a time course experiment. WT seedlings grown under + Pi were transferred to either + Pi (control) or − Pi agar before primary root length was measured at 20 min intervals for 7 h (Fig. 2a). When compared to the control, we detected root growth inhibition 2 h after transfer to − Pi medium, a fast morphological response (Fig. 2b). In contrast, the main root of *stop1*, *almt1* and *lpr1;lpr2* mutants did not respond during the entire 7.5 h time course (Fig. 2c). The short primary root length of WT under − Pi correlated with a strong reduction of root epidermal cell length (Fig. 2d). In the mutants, epidermal cell length was not or only slightly reduced under − Pi (Fig. 2d). Our data show that STOP1, ALMT1 and the LPRs mediate a rapid inhibition of root cell elongation that contributes to root growth inhibition on − Pi medium.

**stop1 and almt1 uncouple the − Pi response of the RAM and EZ.** Previous work showed that the root growth arrest under − Pi is a Fe-dependent response[7,8,12]. Inductively coupled plasma mass spectrometry analysis did not show significant differences in total Fe accumulation in roots of mutants compared to WT (Supplementary Fig. 7a). However, using histological Perls/DAB (3,3′-diaminobenzidine) staining[33], it was reported that labile (non-haem) Fe accumulates in Pi-deprived WT but not *lpr1;lpr2* root tips[12,19]. We thus tested whether the *stop1* and *almt1* mutations alter the pattern of Fe accumulation. By using Perls/DAB staining, we confirmed that WT but not *lpr1;lpr2* root tips accumulated Fe in the EZ and SCN on − Pi medium (Fig. 3a and Supplementary Fig. 7b). Interestingly, *stop1* and *almt1* mutants did not show Fe accumulation in the EZ (Fig. 3a and Supplementary Fig. 7b), but, to our surprise, both mutant lines still accumulated Fe in the SCN similar to the WT (Fig. 3a and Supplementary Fig. 7b).

Callose accumulation in the EZ and SCN is another root response of the WT, but not of the *lpr1;lpr2* double mutant, when grown under − Pi (ref. 12; Fig. 3b). Mirroring the Fe accumulation (Fig. 3a), the *stop1* and *almt1* mutants still deposited callose in the SCN but not in the EZ (Fig. 3b). Note that exogenous malate application complemented iron and callose deposition in the EZ of the *almt1* root tip (Supplementary Fig. 8). These unexpected observations indicate that loss of STOP1 or ALMT1 uncouples the two major regions of Fe and callose accumulations in the root apex.

Because inhibition of cell elongation under − Pi is a rapid process (Fig. 2b) and occurs well before the onset of root meristem size reduction[12], we hypothesized that primary root

**Figure 1 | Phenotype of *almt1* and *stop1* mutants and expression of *ALMT1*.** (**a**) Position and nature of the *almt1* and *stop1* mutations. (**b**) Appearance of representative WT and mutant seedlings after germination and growth for 6 days on + Pi or − Pi medium. (**c**) Primary root length of WT (Col[er105]) and mutant seedlings grown for 6 days in + Pi or − Pi conditions. Mean ± s.d., n = 9–14 seedlings per line and condition; unpaired two-tailed *t*-test; ****P < 0.0001. (**d**) Malate complementation. WT and *almt1* seedlings grown under + Pi for 2 days were transferred for 6 days to − Pi medium with or without the indicated concentrations of malate (mean ± s.d., n = 7–14 seedlings per condition; two-way ANOVA, ****P < 0.0001; ***P < 0.001; **P < 0.01; *P < 0.05; NS, not significant). (**e**) Gene expression analysis. WT seedlings grown under + Pi for 5 days were transferred to + Pi or − Pi medium for 1–48 h, total root RNA was extracted and expression of the indicated genes analysed by qRT–PCR. *SPX1* was used as a positive marker for the − Pi stress. Relative expression levels are normalized to that of the + Pi at 1 h (mean ± s.e.m., n = 6 independent experiments; unpaired two-tailed *t*-test; **P < 0.01; *P < 0.05; NS, not significant). (**f**) Pattern of *ALMT1* expression in primary root tip. Transverse (top) and longitudinal (bottom) sections of the root tip of *pALMT1::GUS* (WT). Seedlings were grown 5 days under + Pi and transferred to + Pi or − Pi medium for 48 h prior to GUS staining. (**g**) GFP-fluorescence of GFP-STOP1 in root tips. Four-day-old *stop1*[KO];*pSTOP1::GFP-STOP1* (# B10A) seedlings grown under + Pi were transferred to + Pi or − Pi plates for 24 h, and GFP-fluorescence was visualized by confocal microscopy. Left panel: the two roots were mounted side by side on the same microscope slide ( × 10 objective). Middle panel: magnification of the root tips ( × 20 objective). See also Supplementary Fig. 6c for additional lines. Right panel: intensity of the GFP-fluorescence (a.u., arbitrary units) in nuclei of the root tip (mean ± s.d., n = 54 and 43 nuclei in + Pi and − Pi, respectively; unpaired two-tailed *t*-test; ****P < 0.0001. Two (**c,d,f**) or three (**g**) independent experiments were performed with consistent results and one representative experiment is shown. Scale bars, 1 cm (**b**), 100 μm (**f,g**).

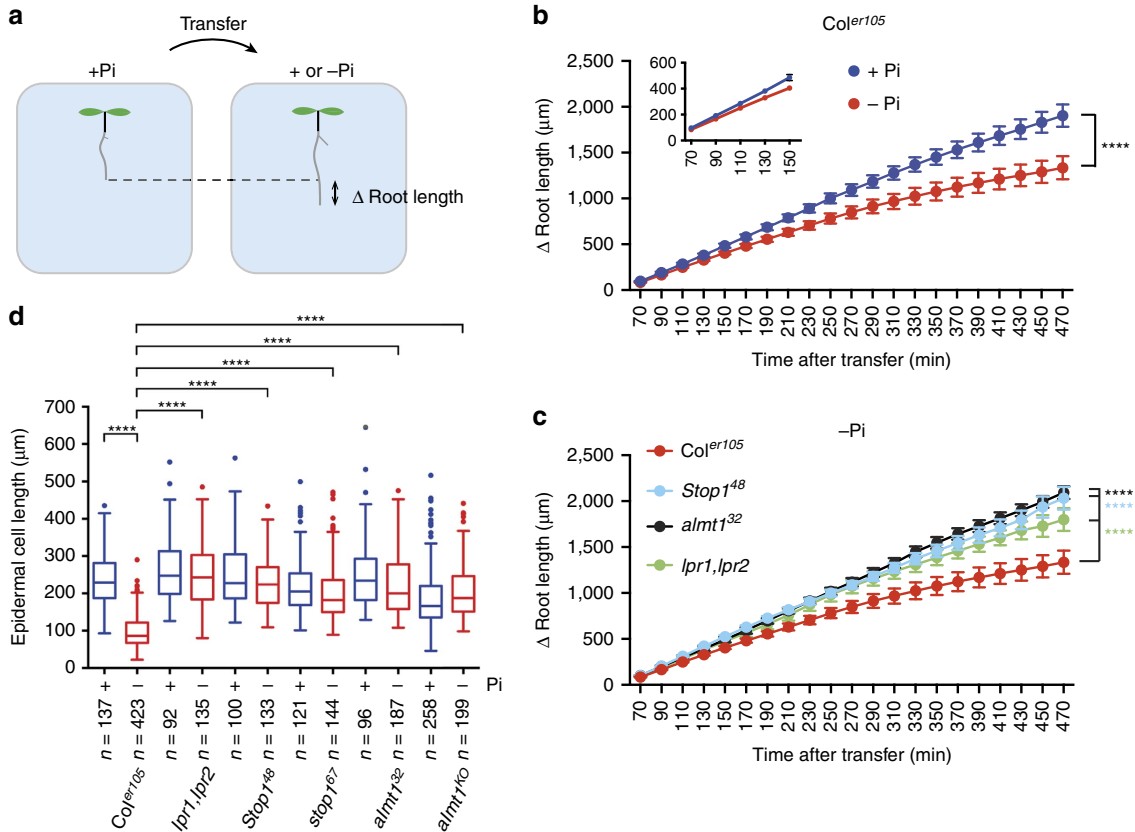

**Figure 2 | Rapid primary root growth arrest under − Pi.** (**a**) Scheme depicting the Δ primary root length measured in seedlings transferred from a + Pi plate to a + Pi or − Pi medium. (**b**) Time course of the Δ primary root length of WT seedlings transferred to + Pi or − Pi medium. Inset shows magnification of the first time points. Three independent experiments were performed with consistent results and one representative experiment is shown (mean ± s.e.m., $n = 10$ seedlings per condition; two-way ANOVA, multiple comparisons; ****$P < 0.0001$). (**c**) Time course of the Δ primary root length of WT and mutant seedlings transferred to − Pi. Two independent experiments were performed with consistent results and one representative experiment is shown (mean ± s.e.m., $n = 8$-10 seedlings per line and condition; two-way ANOVA, multiple comparisons; ****$P < 0.0001$). (**d**) Seven-day-old seedlings of the indicated genotype were transferred to + Pi or − Pi medium for 24 h before measuring the final length of root epidermal cells (median ± interquartile; Tukey whiskers; Mann–Whitney test: ****$P < 0.0001$; $n$, number of cells).

growth inhibition of *stop1* and *almt1* mutants will be detected only after long-term exposure to − Pi. Indeed, when grown for more than 4 days on − Pi medium, the root length of *stop1* and *almt1* mutants was shorter when compared to *lpr1* and *lpr1;lpr2* roots (Fig. 3c), which correlated with a shorter RAM, a proxy of cell proliferation activity (Fig. 3d). Together with root length and growth kinetics analysis, our results show that the absence of rapid inhibition of root growth of *stop1* and *almt1* under − Pi correlates with reduced accumulation of Fe and callose in the EZ, whereas the late root growth inhibition correlates with accumulation of Fe and callose in the SCN and reduced RAM size. Thus, inactivation of the STOP1−ALMT1 module uncouples the effect of − Pi on root cell elongation and proliferation, whereas loss of LPR function impacts both processes (Fig. 3e).

**− Pi stress rapidly stiffens CW in the transition zone.** We next asked how − Pi inhibits cell elongation. In plants, one mechanism that could alter cell expansion is the modification of the mechanical properties of their surrounding CW[34]. We reasoned that CW stiffness increases early after the onset of − Pi stress to prevent expansion of the encased root cell symplast. To test this hypothesis, we used atomic force microscopy (AFM) with a microindentation probe, which allows measuring stiffness of small plant cell surfaces in living organs[35,36]. We measured

the stiffness of the root surface of the transition zone (located between the RAM and EZ, Supplementary Fig. 1a), where cells are predestined to rapidly elongate when conditions are permissive.

AFM measurements were performed on WT seedlings on transfer from + Pi to + Pi or − Pi for up to 2 h (see Methods). Interestingly, CW stiffness increased as early as 30 min after transfer to − Pi and continued to increase at later time points (Fig. 4a); by contrast, we observed a reduced stiffness under + Pi (Fig. 4a). A higher CW stiffness (see also WT in Fig. 4b) was not detectable for the *stop1*, *almt1* and *lpr1;lpr2* lines (Fig. 4b) in which root cell expansion is not restrained under − Pi (Fig. 2d), nor in WT grown in − Pi-Fe condition during which root growth is not inhibited[8,12] (Fig. 4c). Thus, our results reveal a negative reciprocal relation between CW stiffness in the root transition zone and the final epidermal cell length. This suggests a simple causal scenario where − Pi triggers CW stiffening of pre-elongated cells to restrict their elongation.

**Peroxidase activity increases CW stiffening under − Pi.** Class III peroxidases are thought to inhibit cell expansion by catalysing crosslinks between some polysaccharides or proteins, thereby tightening the CW[34,37]. The *Arabidopsis* genome contains 73 genes of class III peroxidases[38]; many of which are expressed

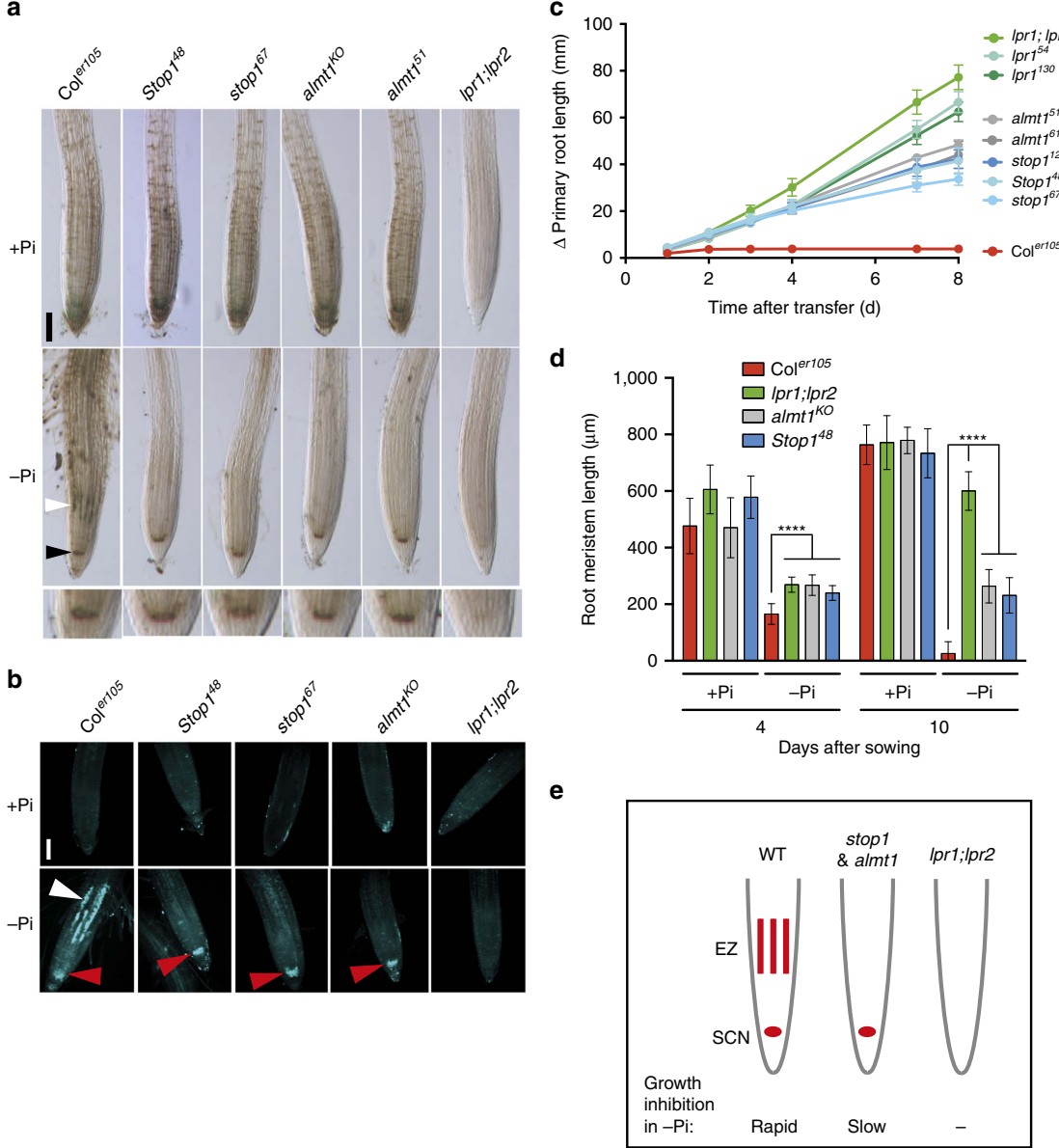

**Figure 3 | The *stop1* and *almt1* mutations largely uncouple the effects of − Pi on the EZ and RAM.** (**a**) Fe accumulation and distribution in primary root tips. Three-day-old seedlings of the indicated genotypes were transferred to + Pi or − Pi plates for 48 h prior to Perls/DAB staining. Note the accumulation of Fe in the EZ of the WT (white arrowhead) and in the SCN (black arrowhead). Bottom panel: magnification of the root tip of seedlings under − Pi. Images are representative of three independent experiments. Scale bar, 100 μm. (**b**) Four-day-old seedlings grown under + Pi were transferred 20 h in + Pi or − Pi before callose staining. The white arrowhead points to callose accumulation in the EZ; the red arrowheads point to callose accumulation in the SCN. Two independent experiments were performed with consistent results and one representative experiment is shown. Scale bar, 100 μm. (**c**) Time course of the Δ primary root length of WT and mutant seedlings transferred to − Pi. Two independent experiments were performed with consistent results and one representative experiment is shown (mean ± s.d., $n = 8–14$ seedlings per line). (**d**) Primary root meristem length of WT and mutant seedlings grown for 4 or 10 days in + Pi or − Pi medium (mean ± s.d., $n = 14–25$ meristems per condition; two-way ANOVA, multiple comparisons; ****$P < 0.0001$). (**e**) Schematic diagram of the consequences of *stop1*, *almt1* and *lpr1;lpr2* mutations on accumulation of Fe and callose, and growth of the primary root under − Pi. The WT root (left) accumulates Fe and callose in the EZ and SCN (red strips and dots, respectively) and its growth is rapidly inhibited. In the *lpr1;lpr2* mutant (right), there is no accumulation of Fe and callose and no inhibition of root growth. In the *stop1* and *almt1* mutants (middle), Fe and callose accumulates only in the SCN and the root growth inhibition is slow.

in root tissues[39,40]. To test our hypothesis that the − Pi cue leads to CW stiffening in root tips, we used a pharmacological approach to inhibit peroxidase activity and to overcome a presumed functional redundancy among these enzymes. Salicylhydroxamic acid (SHAM) and methimazole are two peroxidase inhibitors[41,42] and DAB is a classical peroxidase substrate[43], which we used as a competitive inhibitor. These three chemicals restored the WT root growth under − Pi in a dose-dependent manner (Fig. 5a–d). Consistent with our hypothesis, SHAM strongly decreased CW stiffness in − Pi-challenged WT (Fig. 5e), and SHAM as well as DAB treatment significantly increased root epidermal cell length (Fig. 5f).

These results prompted us to histologically visualize peroxidase activity in the root apex of the WT and mutant lines using

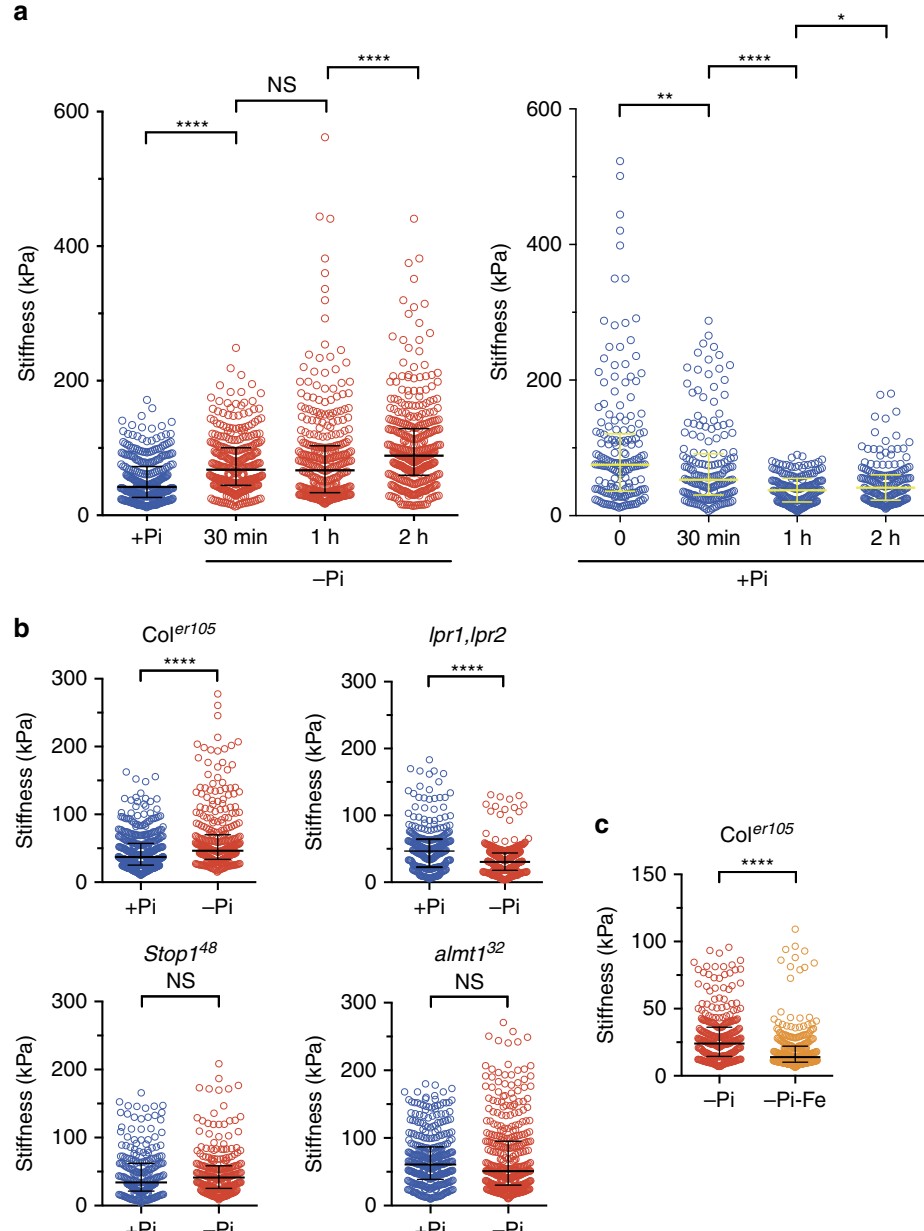

**Figure 4 | −Pi induces cell wall stiffening in the root transition zone.** (**a**) WT (Col$^{er105}$) seedlings were transferred to −Pi or +Pi medium for the indicated time prior to measuring the stiffness of the cell surface by AFM in the transition zone of the primary root (see Methods). (**b**) Seedlings of the indicated genotypes were transferred to +Pi or −Pi medium for 30 min prior to measuring the stiffness of the cell surface by AFM in the transition zone of the primary root. (**c**) WT (Col$^{er105}$) seedlings were transferred to −Pi or −Pi-Fe medium for 30 min prior to measuring the stiffness of the cell surface by AFM in the transition zone of the primary root (median ± interquartile, Mann–Whitney test. ****$P < 0.0001$; **$P < 0.01$; *$P < 0.05$; NS, not significant ($P > 0.05$)). The experiment was performed twice with consistent results and one representative experiment is shown.

4-chloro-1-naphtol staining. We detected peroxidase activity in the WT root apex, from the root cap to the early EZ (Fig. 5g). Supporting our hypothesis, peroxidase staining was more intense in Pi-deprived than Pi-replete WT root tips (Fig. 5g). SHAM-treated seedlings displayed reduced staining, thus confirming SHAM-dependent inhibition of peroxidase activity *in planta* (Fig. 5g and Supplementary Fig. 9a). Remarkably, in the *lpr1;lpr2* double mutant, peroxidase activity was strongly reduced in both +Pi and −Pi conditions, whereas the *stop1* and *almt1* mutants displayed intermediate activities (Fig. 5h). Note, when compared to the WT in both +Pi and −Pi conditions, the stained zone of the *stop1* root tips was shorter and particularly weaker in the EZ (Fig. 5h).

To further challenge the idea that high peroxidase activity correlates with reduced root growth under −Pi, we performed peroxidase staining on WT seedlings grown at pH 7.1 or in a Fe-deficient medium, which are two permissive conditions for root growth under −Pi (ref. 8). As shown (Supplementary Fig. 9b,c) in either condition, peroxidase activity of WT root tips was highly reduced and indistinguishable to that of the mutants. Note that at pH 7.1 no *ALMT1* expression was detected (Supplementary Fig. 4a). These results convincingly show that under −Pi, LPR, and to lesser extent STOP1 and ALMT1, stimulate peroxidase activity in the root apex, which in turn inhibits root growth. Combined with our AFM and

pharmacological results, these observations support the view that peroxidase activity catalyses crosslinks in some CW components, thereby reducing cell expansion under − Pi.

It is worth noting that the CW stiffness of SHAM-treated WT (Fig. 5e), as well as of WT grown under − Pi-Fe (Fig. 4c) and in *lpr1;lpr2* double mutant grown under − Pi (Fig. 4b), is lower than in the corresponding controls grown under + Pi. This softening is correlated with reduced peroxidase activity observed in these roots (Fig. 5g, Supplementary Fig. 9b and Fig. 5h). These observations may indicate that in conditions permissive for root

growth, the LPR ferroxidases and some peroxidases participate to maintain a basal strength of the CW.

Under − Pi, peroxidase activity also represses cell proliferation and accelerates cell differentiation in the RAM (Fig. 5d, pictures of the root tips). This is reminiscent to the role of UPB1, which regulates the transition from cell proliferation to differentiation in the root tip via the expression of some class III peroxidases[44]. However, primary root growth of the *upb1* mutant and an UPB1-overexpressor line is not altered in response to − Pi exposure (Supplementary Fig. 9d), indicating that UPB1 is not essential in this process.

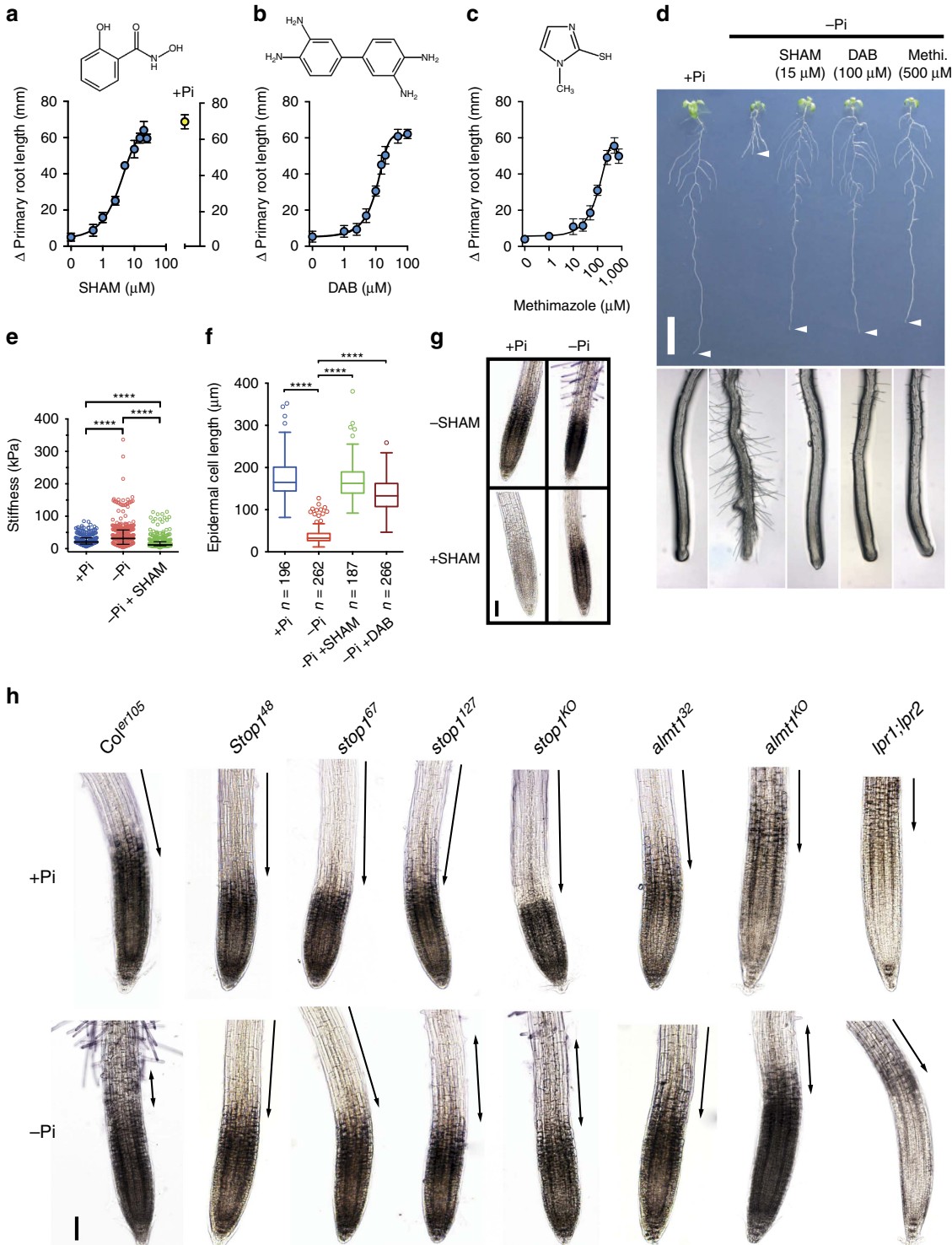

**− Pi promotes ROS accumulation in the EZ.** Correlated with the deposition of Fe and callose, the generation of ROS in CWs of the RAM and EZ was observed in WT, but not in the *lpr1;lpr2* double mutant, when transferred to − Pi medium for 1 day[12,19]. We also observed accumulation of ROS ($H_2O_2$ and peroxidases-mediated oxidation of CM-H2DCFDA) in the transition and differentiation zones of WT seedlings transferred to − Pi for 24 h (Supplementary Fig. 10). Under − Pi, accumulation of ROS is reduced when iron is omitted or when SHAM is added to the growth medium (Supplementary Fig. 10). This result confirms that − Pi induces accumulation of ROS in the root tip and indicates that peroxidase activity could play a role.

**Malate effluxed into the RAM inhibits root growth under − Pi.** Specific cell types of the RAM control root growth in *Arabidopsis* (see for example ref. 45). To test whether STOP1 activity in specific cell files inhibits root growth under − Pi, we used cell type-specific promoters to confine and restore *STOP1* expression (and thus *ALMT1* expression) in a *stop1*$^{KO}$ mutant (Fig. 6a). As shown in Fig. 6b, we fully rescued root growth inhibition by transformation with a *pSCR::STOP1* construct, which drives *STOP1* expression specifically in the QC and endodermal cell lineage[46]. The *pCO2* promoter[46] restored *STOP1* expression essentially in young cortex cells of the RAM and complemented *stop1*$^{KO}$ almost fully. The deposition of Fe in the EZ was also complemented in the *pSCR::STOP1* and *pSCR::Co2* lines (Fig. 6c).

In contrast, we observed only partial complementation using the *pPEP* promoter, which is active in cortex cells of the elongation and maturation zones[47] and slightly in the endodermis[47], and no complementation with the *pCASP1* promoter, which is active in mature endodermal cells[48] (Fig. 6b). Using a *pALMT1::GUS* reporter, the expected pattern where STOP1 is functional in the root and root tip of all these lines was confirmed (Supplementary Fig. 11a,b). Taken together, our results show that *STOP1* expression confined to the endodermis (and QC) or to the early cortex is sufficient for rescuing the *stop1*$^{KO}$ root phenotype.

To test whether these two cell types are necessary for STOP1-dependent growth arrest, we generated WT lines expressing the dominant negative *Stop1*$^{48}$ gene under the *pSCR* or *pCo2* promoter. Our results show that the two constructs, expressed either alone (Supplementary Fig. 11c) or in combination (Fig. 6d), do not release root growth inhibition under − Pi.

Taken together, these experiments show that expression of *STOP1* in endodermis or cortex of root tips is sufficient, but not necessary, for STOP1-dependent root growth inhibition by − Pi. This indicates that ALMT1-exported malate locally diffuses in the apoplast of root tip cells.

## Discussion

Identification of the cellular and molecular mechanisms by which environmental cues affect organ growth is a current challenge for plant biology. Pi starvation is one of the most common limiting factors for plant growth and crop productivity. Previous studies identified *Arabidopsis* LPRs and PDR2 as part of a pathway controlling primary root growth in response to Pi availability surrounding the root tip, the so-called 'local response'[10,11].

Our work uncovers a novel pathway in local Pi sensing. We identified in a forward genetic screen two functionally interacting components with roles in primary root growth arrest under − Pi: the transcription factor STOP1 and its direct target gene *ALMT1*. Absence of a reciprocal regulation of *pALMT1* and *pLPR1* promoter activity (Supplementary Fig. 12a,b) suggests that STOP1–ALMT1 and LPR1 function in distinct pathways. However, the similar unrestricted root cell elongation in *stop1*, *almt1* and *lpr1* mutants (Fig. 2c,d) indicates that the two pathways converge on a common inhibition mechanism. Therefore, the PDR2–LPR1 and STOP1–ALMT1 modules define the first components of two separate pathways targeting the root transition zone to inhibit cell elongation under − Pi (Fig. 7).

To our knowledge, STOP1 is the first plant transcription factor showing a regulation reminiscent of yeast PHO4 (ref. 49), which accumulates more intensely in the nucleus upon Pi starvation. Thus, STOP1 and ALMT1 define a new − Pi regulatory pathway, which is supported by PHR1/PHL1-independent activation of *ALMT1* (Supplementary Fig. 3a). In addition to low pH and $Al^{3+}$ (refs 21,22,24,32), low Pi is thus a new environmental stimulus activating the STOP1–ALMT1 pathway in *Arabidopsis* and most probably in crops[50]. Indeed, the primary root of the *Brassica napus*, a crop plant known to exude malate under − Pi (ref. 51), is also inhibited under − Pi (Supplementary Fig. 13). This pathway is thus likely of general relevance.

Our work further uncovers that root growth inhibition under − Pi results from rapid inhibition ($<2$ h) of cell elongation and subsequent inhibition ($<4$ days) of cell proliferation, previously referred to as meristem exhaustion[5,9,12,52,53]. Interestingly, our data demonstrate that both processes can be genetically uncoupled in *stop1* and *almt1* mutants. Both inhibitory processes correlate with a higher Fe accumulation, callose deposition, and peroxidase activity in the EZ and SCN, as well as with CW stiffening at least in the EZ. Recently, a comparative

**Figure 5 | Cell wall stiffening and inhibition of root cell elongation under − Pi are peroxidase-dependent.** (**a–c**) Dose–response relationship of the Δ primary root length plotted as a function of concentration of peroxidase inhibitors (mean ± s.d.). For each data point, 3-day-old WT (Col$^{er105}$) seedlings were transferred for 7 days to − Pi plates containing different concentrations of inhibitor. The structure of the inhibitors is shown above the corresponding curves. (**a**) Salicylhydroxamic acid (SHAM). Blue data points: − Pi; yellow data point, + Pi without inhibitor ($n = 12–14$ seedlings per condition). (**b**) 3,3′-diaminobenzidine (DAB) ($n = 8–12$ seedlings per condition). (**c**) Methimazole ($n = 11–13$ seedlings per condition). (**d**) Picture of the effect of peroxidases inhibitors on WT seedlings. Three-day-old WT seedlings were transferred for 7 days to + Pi or − Pi plates containing the indicated inhibitors. Top: picture of one representative seedling for each treatment (Methi., methimazole). Arrowheads indicate the primary root tips. Scale bar, 1 cm. Bottom: pictures of the root tips corresponding to the seedlings shown above. Note that inhibitors of peroxidases suppress the − Pi-induced early differentiation of root hairs. (**e**) Three-day-old WT seedlings were transferred for 30 min to + Pi or − Pi medium with or without 15 μM SHAM prior to measuring the stiffness of the cell surface by AFM in the transition zone of the primary root (median ± interquartile, Tukey whiskers, Mann–Whitney test: \*\*\*\*$P < 0.0001$). (**f**) Effect of SHAM and DAB on root epidermal cell length. Three-day-old WT seedlings were transferred for 7 days to + Pi or − Pi medium with or without 15 μM SHAM or 100 μM DAB prior to measuring the root epidermal cell length (median ± interquartile; Tukey whiskers, Mann–Whitney test: \*\*\*\*$P < 0.0001$, $n =$ number of cells). (**g**) Peroxidase activity revealed by 4-chloro-1-naphtol staining of the WT primary root tip (the colourless dye turns black with peroxidases activity). Four-day-old WT seedlings were transferred for 48 h to + Pi or in − Pi plates with or without 15 μM SHAM (see also Supplementary Fig. 9a). (**h**) Peroxidase activity revealed by 4-chloro-1-naphtol staining of the primary root tip of the indicated genotypes. Four-day-old seedlings were transferred for 48 h to + Pi or in − Pi plates prior to staining. Single and double-headed arrows indicate the position of the elongation zone. The experiments were performed three times with consistent results and one representative experiment is shown (**a–c,g,h**). Scale bars, 100 μm (**g,h**).

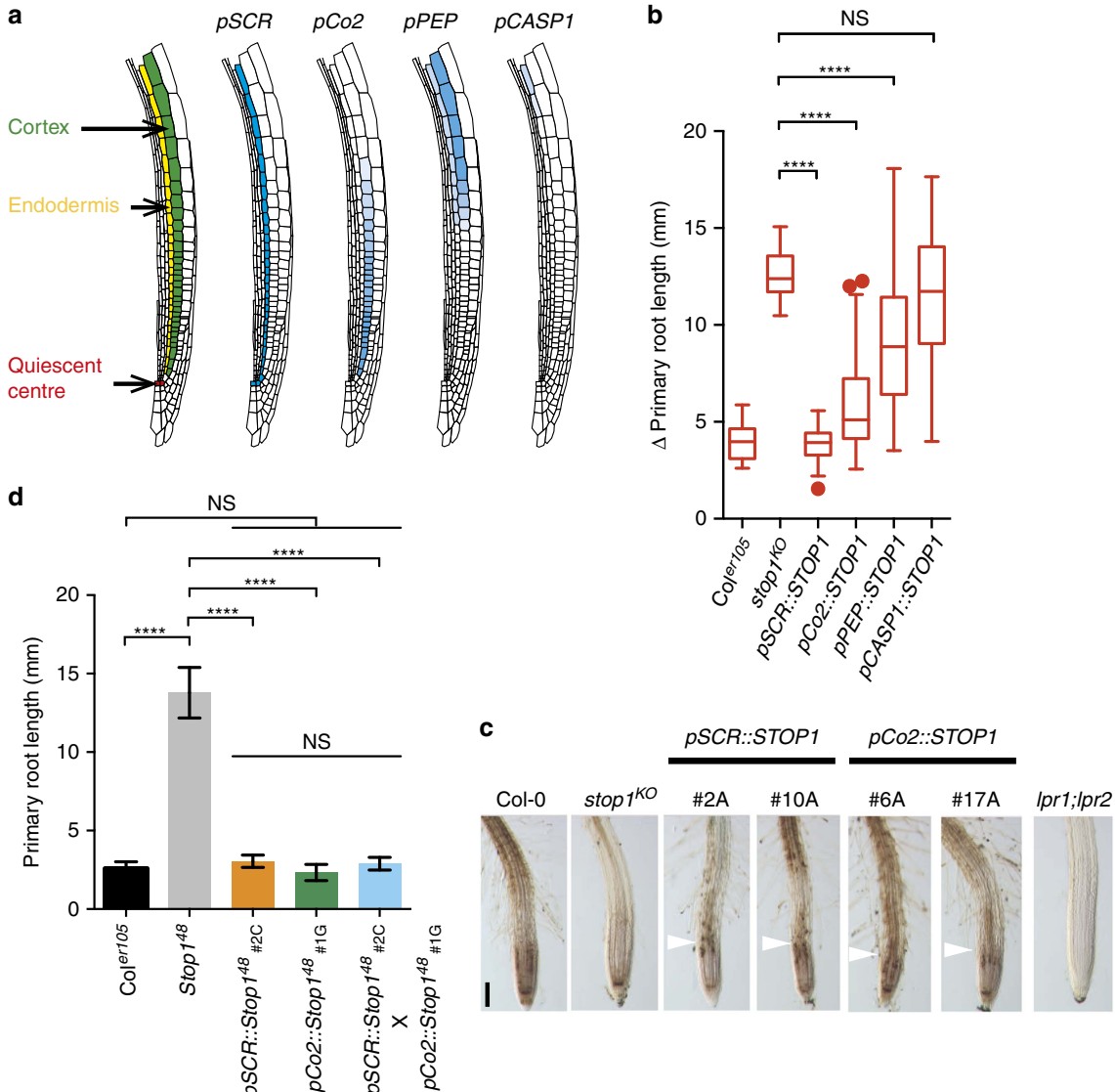

**Figure 6 | Expressing *STOP1* in the endodermis or early cortex of the *stop1* mutant is sufficient to restore root growth arrest under −Pi. (a)** Schematic diagram showing the position of cortex, endodermis and quiescent centre, and the tissue-specific expression (blue) conferred by the indicated promoters. Only one longitudinal half of the root is shown. **(b)** Primary root growth of *stop1$^{KO}$* mutant lines with tissue-specific expression of *STOP1*. Three-day-old seedlings were transferred for 4 days to −Pi and the Δ primary root length measured in 5–6 independent transgenic lines par construct (mean ± s.d., *n* = 17 seedlings per line; box and Tukey whiskers; statistics by unpaired two-tailed *t*-test: ****$P < 0.0001$; NS, not significant). The experiment was performed twice with consistent results; one experiment is shown. **(c)** Fe accumulation and distribution in primary root tips of *pSCR::STOP1* and *pCo2::STOP1* lines. Three-day-old seedlings of the indicated genotypes (two independent lines for each construct) were transferred from +Pi to +Pi or −Pi plates for 48 h prior to Perls/DAB staining; one representative seedling is shown. Note the accumulation of Fe in the EZ of the complemented lines (white arrowhead) compared to the parental *stop1$^{KO}$* line. Scale bar, 100 μm. **(d)** The dominant negative *Stop1$^{48}$* was expressed in the endodermis (*pSCR::Stop1$^{48}$*), in the cortex (*pCo2::Stop1$^{48}$*) or in both cell types (F1 progeny from the cross *pSCR::Stop1$^{48}$* X *pCo2::Stop1$^{48}$*) in Col$^{er105}$. Seedlings were grown for 5 days under −Pi and the primary root length measured (mean ± s.d., *n* = 7–14 seedlings per line; one-way ANOVA; ****$P < 0.0001$; NS, not significant). Note that the primary root of transgenic lines behaves like in WT control.

transcriptomics/proteomics study revealed that Pi deprivation profoundly alters expression of CW-related genes (<20 h) and leads to accumulation of callose and non-esterified pectins in WT root tips at sites of Fe deposition (EZ, SCN), which are not detected in *lpr1;lpr2* roots[12,19]. Crosslinking of non-esterified pectins by $Ca^{2+}$ reduces CW extensibility and regulates cell expansion[54].

It is quite unexpected that malate efflux inhibits elongation of root cells under −Pi, because exudation of small organic acids is thought to promote plant growth by solubilizing scarcely available Pi and Fe nutrients, or by forming non-toxic complexes

with $Al^{3+}$ cations[25,26,55,56]. However, malate and Fe form chelates participating in plant Fe homeostasis[57]. Interestingly, when complexed with malate, $Fe^{3+}$ reduction is faster when compared to $Fe^{3+}$–citrate or $Fe^{3+}$–EDTA complexes[58]. Therefore, a low Pi/Fe ratio, as under −Pi, may allow formation of $Fe^{3+}$–malate complexes in CWs of the RAM and EZ, promoting Fe accumulation, LPR1 ferroxidase-dependent Fe redox cycling, ROS generation and callose accumulation (Fig. 7).

The role of Fe might not be limited to ROS production; one might speculate that $Fe^{3+}$ also activates ALMT1 to control malate efflux, as $Al^{3+}$ regulates ALMT1 channels from different

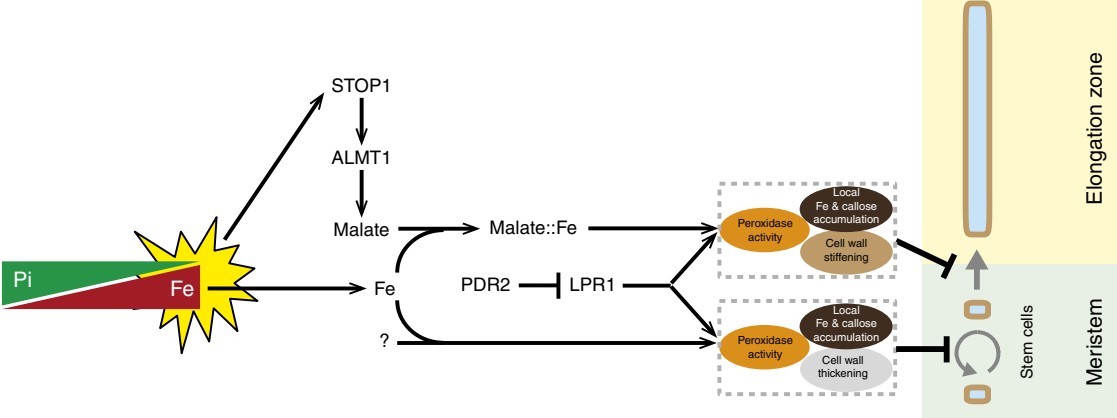

**Figure 7 | Working model depicting the effect of $-$Pi on cell proliferation and elongation in the primary root tip.** A low Pi/Fe ratio, that is, $-$Pi stress[12] (yellow star), post-transcriptionally activates STOP1 which induces *ALMT1* expression followed by malate exudation into the apoplast. The interaction of malate-altered iron chelation and LPR1 ferroxidase activity inhibits cell elongation by peroxidase-dependent cell wall stiffening. The P5-type ATPase PDR2 restricts LPR1 function[12]. The LPR1–PDR2 module interacts with an unknown pathway parallel to STOP1–ALMT1 (question mark) to inhibit cell division in the SCN by a similar mechanism involving accumulation of Fe and callose, peroxidase-activity, and cell wall thickening. Mutations in *STOP1* and *ALMT1* restore cell elongation under $-$Pi but cell divisions are still inhibited.

species[20,59–63]. However, the latter mechanism is still debated for AtALMT1 (ref. 64). Gamma-aminobutyric acid, pH and external anions also regulates the activity of wheat ALMT1 (ref. 62) and we do not exclude the possibility that small molecules also stimulate AtALMT1 activity under $-$Pi.

As shown at pH 7.1 (Supplementary Fig. 4a), *ALMT1* is not expressed in $+$Pi and $-$Pi conditions (maybe because the STOP1 protein is not activated at this pH). This could partly explain the lack of root growth inhibition under $-$Pi at this pH (ref. 8). However, even when *ALMT1* is constitutively expressed, there is no growth inhibition (Supplementary Fig. 4b); one explanation might be that at higher pH Fe precipitates in a form that cannot inhibit root growth.

Peroxidases produce or consume ROS[37]. Restoring root growth under $-$Pi with the competitive artificial substrate DAB (Fig. 5b,d) supports the view that peroxidases inhibit growth via their peroxidative activity that consumes $H_2O_2$. This activity of class III peroxidases is known to catalyse intermolecular covalent bonds between CW components like monolignols, feruloylated pectins or structural proteins such as hydroxyproline-rich glycoprotein and extensins[34,37]. These crosslinks are thought to impede CW expansion and therefore prevent cell enlargement. The increased (early) CW stiffness was confirmed by nanoindentation experiments (AFM), which fully support a causal link between peroxidase activity and the arrest of cell elongation. Likewise, it was recently reported that Pi deprivation deregulates the expression of about 30 class III peroxidases at the mRNA or protein level[19] and causes increased lignification in WT but not in *lpr1;lpr2* root tips[65].

Interestingly, peroxidase-mediated strengthening of extracellular fibrillar proteins (for example, collagen) by crosslinking also operates in animals, pointing to an evolutionary conserved mechanism[66,67].

In conclusion, we propose a new pathway and mechanism for local Pi sensing by root tips. Pi limitations enhance the nuclear abundance of STOP1 to transcriptionally activate in a PHR1-independent manner *ALMT1* expression, which promotes malate efflux by internal cell layers of the root apex. Malate-facilitated Fe chelation and LPR1-dependent Fe redox cycling stimulate both the peroxidase-dependent CW stiffening, which rapidly inhibits root cell elongation, and the accumulation of callose in the EZ (Fig. 7). With its dual role in promoting or attenuating root growth, exuded malate may be considered a genetically controlled intercellular diffusible factor that adjusts root cell elongation to the local chemical environment of the rhizosphere.

## Methods

**Plant growth.** Unless specified, plants and seedlings were grown as previously described[68]. Briefly, surface-sterilized seeds were sown on a horizontal line in plates that were vertically disposed in a growing chamber (16 h photoperiod; 24.5 °C/ 21 °C day/night, respectively). The growth medium contained 0.15 mM $MgSO_4$, 2.1 mM $NH_4NO_3$, 1.9 mM $KNO_3$, 0.34 mM $CaCl_2$, 0.5 μM KI, 10 μM $FeCl_2$, 10 μM $H_3BO_3$, 10 μM $MnSO_4$, 3 μM $ZnSO_4$, 0.1 μM $CuSO_4$, 0.1 μM $CoCl_2$, 0.1 μM $Na_2MoO_4$, 0.5 g l$^{-1}$ sucrose. The agar (0.8 g l$^{-1}$) for plates was from Sigma-Aldrich (A1296 #BCBL6182V). The $-$Pi and $+$Pi agar medium contained 10 and 500 μM Pi, respectively; the media were buffered at pH 5.6–5.8 with 3.4 mM 2-(N-morpholino)ethane sulfonic acid.

Growth media at pH 7.1 were made using 2 mM 1,4-piperazinediethanesulfonic acid pH 7.1. Growth media at pH 4.4 were prepared by mixing equal amount of autoclaved 2× liquid medium (composition as above, without Pi and Fe, buffered at pH 4.4) with 2× agar (dissolved in $H_2O$); Pi and Fe were subsequently added according to media. For the pH experiment (Supplementary Fig. 4a), Fe was at 15 μM.

*Brassica napus* seeds (cv Bohème) were sterilized and seedlings grown on the same media, as for *Arabidopsis*, except that the growth media contained 100 μM $FeCl_2$. Briefly, seeds were sown on a $+$Pi plate, left 24 h at 4 °C and let germinate for 12 h. Germinating seeds (with the primary root just protruding) were then transferred in $+$Pi or $-$Pi plates for 3 days before measuring the primary root length.

For testing resistance to $Al^{3+}$, seeds were sown in 2 ml liquid medium ($NH_4NO_3$ 4.1 mM, $KNO_3$ 3.8 mM, $CaCl_2$ 200 mM, $MgSO_4$ 0.3 mM, $H_3BO_3$ 20 μM, $MnSO_4$ 20 μM, KI 1 μM, $ZnSO_4$ 6 μM, $Na_2$ $MoO_4$ 0.2 μM, $CoCl_2$ 0.2 μM, $CuSO_4$ 0.2 μM, sucrose 0.5 g l$^{-1}$, Homopiperazine-1,4-bis(2-ethanesulfonic acid) 2 mM pH 4) and grown for 4 days before measuring the primary root length.

**Mutagenesis and mutants screening.** For the mutagenesis, we used Col$^{er105}$, a Columbia background (Col-0) with the null allele *erecta-105* (ref. 69) that confers a stunted habitus facilitating handling of adult plants in the growth chambers.

About 8 × 10$^4$ Col$^{er105}$ seeds were mutagenized with 0,2% of ethylmethane sulfonate (EMS, Sigma-Aldrich) for 16 h, under gentle agitation. The EMS was inactivated with 100 mM sodium thiosulfate for 10 min followed by washing the seeds with tap water (10 times). Seeds were sown on soil and the M2 generation was collected in 156 pools of ~400–500 M1 plants each. From each pool, about 2,500 seeds were sown on two plates (12 × 12 cm) containing $-$Pi medium (each plate contained two horizontal lines of about 625 seeds each). After 12 h at 4 °C plates were placed on a 30° vertical angle in a growth chamber. After 7 days, plates were screened for seedlings with a long primary root (in comparison to neighbouring roots), and putative mutants were transplanted into soil. The mutant phenotype was checked in the M3 generation and backcrossed to the WT Col$^{er105}$ line. When several putative mutants appeared in a pool, only one was kept for further study. From a total of about 390,000 M2 seeds sown on $-$Pi medium,

we isolated 85 viable putative mutants with a long primary root. The mutant phenotype was confirmed in the M3 generation.

**Mapping $almt1^{32}$ and $Stop1^{48}$.** Two non-allelic mutants (ems32 and ems48), that do not carry mutations *LPR1* (see below), were chosen to identify the corresponding mutant genes by map-based cloning approaches.

The $almt1^{32}$ mutant was crossed with the Sha accession, which has a short root when grown under $-$ Pi (ref. 6). In the F2 population grown under $-$ Pi, we selected seedlings with a mutant phenotype. DNA of individual lines was extracted, and a combination of high-resolution melting (HRM) and microsatellites molecular markers were tested for genetic linkage analysis. The mutation was mapped between molecular markers MSAT1–2.62 and PMSha-2.78. For the fine mapping, we developed new microsatellite and HRM markers (Supplementary Table 1), allowing localization of $almt1^{32}$ between MSAT1–2.62 and HRM2745575 markers, a 120 kb interval.

The same procedure was applied to $Stop1^{48}$ except that we selected F2 seedlings with a WT phenotype (because $Stop1^{48}$ is dominant). The locus was mapped between MSAT1–12.22 and CWI1 in a 6 Mb interval. Among $\sim$1,440 chromosomes, 7 and 107 were recombinant for MSAT1–12.22 and for CWI1, respectively.

**Identification of the $almt1^{32}$ and $Stop1^{48}$ mutations.** The recessive ems32 and the dominant ems48 mutations were mapped on the top arm of chromosome I in regions of 120 kb and $\sim$2 Mb, containing 31 and $\sim$6–700 genes, respectively. To identify the corresponding mutant genes, we followed a double gene candidate approach reasoning that the ems32 and ems48 mutations could be located in functionally related genes. By looking at the predicted genes in the 120 kb of chromosome interval containing ems32 we found ALMT1 (At1g08430) and in the 2 Mb region containing ems48 we found STOP1 (At1g34370), encoding a putative transcription factor regulating the expression of *ALMT1* (ref. 22). Sequencing *ALMT1* in the ems32 mutant and *STOP1* in the ems48 mutant confirmed our conjecture. Homozygous T-DNA and Ds insertion mutant alleles of these genes conferred the same long root mutant phenotype as the EMS alleles (Fig. 1c) thus demonstrating that *almt1* and *stop1* mutants have a long root under $-$ Pi.

**Allele screening.** To identify the potential *lpr1* alleles in our collection of EMS-mutagenized mutants, we used a molecular test based on the restriction enzyme ENDO1 that cuts PCR products containing a DNA mismatch. The ENDO1 restriction enzyme was used according to instruction manual (SERIAL GENETICS). Briefly, the *LPR1* gene was amplified with primers 23010cDNAF plus 23010R2 and with 23010F4 and RT10R (Supplementary Table 1) in the mutants and in the WT. A 1/1 ratio of mutant and WT amplicons were mixed in a test tube. After denaturation at 95 °C, DNA was allowed to renature by cooling (1 °C min$^{-1}$) to 5 °C. 0.4 unit of ENDO1 reaction mix containing 2.5 μg of renatured DNA was added and the tubes transferred at 37 °C for 20 min. DNA was electrophoretically resolved on a 2% agarose gel and stained with ethidium bromide to detect amplicons partially cut by ENDO1. To identify the mutation, we sequenced the amplicon obtained in an independent PCR reaction with DNA of mutant lines.

To find potential alleles of *almt1* and *stop1* in our collection of EMS-mutagenized mutants, we first tested all the mutants to identify those with a primary root growth hypersensitive to $Al^{3+}$ (ref. 22). Seedlings of these mutants were grown 24 h in liquid medium containing 50 μM $Al^{3+}$ and the expression of *ALMT1*, *MATE* and *STOP1* was measured by qRT–PCR (primers in Supplementary Table 1). We then sequenced the *STOP1* gene of mutants in which the expression of both *MATE* and *ALMT1* was reduced (compared to WT) and we sequenced the *ALMT1* gene of mutants in which the expression of both *STOP1* and *MATE* was not reduced.

**Other *Arabidopsis* lines.** The $stop1^{KO}$ (SALK_114108, NASC reference N666684), $almt1^{KO}$ (SALK_009629, NASC reference N680999). *lpr1;lpr2* (ref. 8), *phr1;phl1* (ref. 29), $wrky46^{KO}$ (SALK_134310C, NASC reference N667392), *pLPR1::GUS* line[12], $almt1^{KO};mate^{KO}$ carrying the *pALMT1::MATE* construct[27], and *upb1-1* and *35S::YFP-UP1* line[44] are in the Col-0 background. The $mate^{KO}$ (SK3573, NASC reference N1000719) is in Col-4 background.

**Malate complementation.** A volume of 200 μl per well of MS/10 medium[8] without FeCl$_2$, KH$_2$PO$_4$ and agar were distributed in a 12-well plate and used to germinate seeds for 48 h (20 seeds per well and 2 wells per line). The plate was placed in the same conditions (light and hygrometry) as agar plates (see above). After 48 h, when the primary root and cotyledons emerge from the seed coat, the medium was replaced by 2 ml of a fresh MS/10 medium containing 10 μM FeCl$_2$ and 10 μM ($-$ Pi) or 500 μM ($+$ Pi) of KH$_2$PO$_4$ and different concentrations of L-malate (0–200 μM) and adjusted at pH 5.5 with KOH. After 60 h, the medium was replaced by fresh medium of identical composition, and seedlings grown for additional 60 h. At the end of this time each seedling was placed on an agar plate to measure the primary root length.

**Chemical stock solutions.** SHAM (Sigma-Aldrich, France) was dissolved at 10 mM in absolute ethanol, DAB (Sigma-Aldrich, France) was dissolved at 100 mM in dimethyl sulfoxide (Sigma-Aldrich, France) and methimazole (Sigma-Aldrich, France) was dissolved at 100 mM in H$_2$O.

**Quantitative RT–PCR.** Total RNA was extracted from whole roots using the RNeasy Plant Mini Kit (Qiagen, France) and treated with the RNase-free DNase Set (Qiagen, France) according to the manufacturer's instructions. Reverse transcription was performed on 400 ng total RNA using the SuperScript VILO DNA Synthesis Kit (Invitrogen). Quantitative PCR (qRT–PCR) was performed on a 480 LightCycler thermocycler (Roche) using the manufacturer's instructions with Light cycler 480sybr green I master (Roche) and with primers listed in Supplementary Table 1. We used *ROC3* (AT2G16600) as a reference gene for normalization.

**Transgene constructs.** To study promoter activity, *promoter::GUS* lines were constructed for *pALMT1* and *pSTOP1*. The 1.7 and 2.1 kb region upstream of the start codon of the *ALMT1* and *STOP1* gene were amplified respectively from WT genomic DNA. The PCR product was then cloned into the pDONR P4-P1R by a BP clonase reaction (LifeTechnologies). GUS was cloned into a pENTR vector and final constructs were obtained by multisite gateway reaction with pB7m24GW.3 destination vector as described elsewhere[70].

GFP-reporter lines using promoter *UBQ* (*UBIQUITIN10*) were constructed for *ALMT1* and *STOP1*. For localization of STOP1 additionally GFP-reporter lines with the native promoter were constructed. pEN-L1-F-L2 and pEN R2-GFP-L3 were used for GFP-fusion constructs by multisite gateway cloning. The 640 bp upstream region of the start codon of *UBQ10* and 2.1 kb upstream region of the *STOP1* gene was amplified from genomic DNA and cloned into the pDONR P4-P1R. The full-length coding region of *ALMT1* and *STOP1* were amplified by PCR and inserted into the pDONR P2R-P3. The final gateway reaction was accomplished with pB7m34GW as destination vector, resulting in *pUBQ10::GFP-STOP1*, *pUBQ10::ALMT1-GFP* and *pSTOP1::GFP-STOP1*. The resulting vectors were transformed into *Arabidopsis* by floral dip method[71]. Primers used for vector constructions are listed in Supplementary Table 1.

Several independent lines segregating in 3:1 (in the T2 generation) for the resistance to the antibiotic carried by transgene were selected for all our constructs.

To introgress *pALMT1::GUS$_{#2}$* in the $Stop1^{48}$ background, pollen of $Stop1^{48}$ was used to cross the *pALMT1::GUS$_{#2}$* WT line and the cross was checked in F1 grown under $-$ Pi (long primary root). F2 seedlings with a long primary root under $-$ Pi were selected and transferred in phosphinothricin (PPT) plates. Several PPT-resistant plants were transferred in soil for selecting the double homozygous (*pALMT1::GUS$_{#2}$/pALMT1::GUS$_{#2}$*; $Stop1^{48}/Stop1^{48}$) by checking the root phenotype under $-$ Pi and the resistance to PPT.

The *pLPR1::GUS* line[12] has been crossed with $Stop1^{48}$ and $stop1^{127}$ mutants and lines homozygous for both the transgene and the *stop1* mutation selected in the progenies.

**Cell type-specific complementation.** For cell type-specific complementation, we used *pCO2*, *pPEP*, *pCASP1* and *pSCR* promoters as described[46–48,72]. Constructs were generated by multisite gateway reaction with pB7m24GW.3 as destination vector for $STOP1^{WT}$ and $Stop1^{48}$. Vectors were then used to transform the $stop1^{KO}$ or the Col$^{er105}$ (containing *pALMT1::GUS$_{#2}$*) background. F2 seedlings with PPT or kanamycin resistance were selected and transferred to soil.

**Measurement of the primary root length.** Pictures of the plates were taken with a camera and root lengths measured using the plugin NeuronJ[73] for the ImageJ software[74]. For the early kinetic analysis, seedlings were grown 7 days under $+$ Pi and then transferred onto a $+$ Pi or a $-$ Pi medium and pictures of the root tip were taken every 20 min for 7.5 h with an Axiozoom-V16 macroscope (Zeiss). $\Delta$ root lengths were measured with Image-J software[74] and a special M macro developed on purpose (Jérôme Mutterer, IBMP, Strasbourg).

**Measurement of root epidermal cell and RAM lengths.** To stain the CWs of epidermal and cortex cells, seedlings were dipped, respectively, 3 min in 15 μM propidium iodide (PI) or 5 min in 30 μM PI, at room temperature, and rinsed twice with H$_2$O.

To measure RAM length, based on the files of cortex cells (from the QC to the first elongated cell), images were collected on a Zeiss LSM780 confocal microscope (Carl Zeiss, France) using a $\times$10 (no immersion) or $\times$20 (water immersion) objective. For PI visualization, the DPSS laser (561 nm) was used. Emitted light was collected from 600 to 675 nm for PI, using the MBS 561 filter.

Images of epidermal cells, in root zones where elongation has ended, were collected with an Axiozoom-V16 macroscope imaging system (mRFP filter, excitation at 572 nm, emission at 629 nm) using the Apotome functionality and processing with Zen blue (Zeiss). Cell lengths were measured with the ImageJ software (Rasband[74]). For each growth condition, cells were measured in three independent zones, along each of the five roots that were stained.

**Inductively coupled plasma mass spectrometry analysis.** Seeds were sown in line on a sterile nylon mesh (1.5 × 12 cm, 100 μm). They were grown for 5 days under + Pi before transfer for 24 h under + Pi or − Pi thanks to nylon mesh. 60 root tips per genotype (≈ 5 mm length) were harvested with Teflon razor blades and tweezers into cryovial tubes.

Samples were mineralized at 80 °C for 3 h using concentrated $HNO_3/H_2O_2$ 75/25 (v/v). Subsequently, they were diluted 10 times by ultrapure water and injected via a peristaltic pump at 400 μl min$^{-1}$ flow rate in the 7,700 × inductively coupled plasma mass spectrometry analysis (ICP/MS) (Agilent). ICP conditions were the following: nebulization gas flow rate 1 l min$^{-1}$, plasma gas flow rate 15 l min$^{-1}$, auxiliary gas flow rate 1 l min$^{-1}$. Power plasma was set to 1,550 W. The collision cell was activated to decrease polyatomic ion interferences using He at 4 ml min$^{-1}$ flow rate. Other conditions were adjusted to both maximize analyte signals and minimize oxide and doubly charged ion formation. Quantification of Fe was performed at $m/z = 54$ and 56 using external standards. Between analyses, the system was rinsed for 3 min by $HNO_3$ 5% a blank was injected to control the absence of any memory effect. Acquisition was performed for 3 s in triplicate.

**Yeast one-hybrid assay.** The *Saccharomyces cerevisiae* strains used were YPH501 (diploid, *ura3–52, lys2–801, ade2–101, trp1-Δ 63, his3-Δ 200, leu2-Δ 1*) and YPH499 (*MATa, ura3–52, lys2–801, ade2–101, trp1-Δ 63, his3-Δ 200, leu2-Δ 1*).

For yeast transformation, cells were grown in the YPD (yeast extract–peptone–dextrose) medium, and for the yeast one-hybrid assay, cells were grown in raffinose synthetic complete medium minus uracil and tryptophan, supplemented with 1% of galactose synthetic complete medium minus uracile and trytophan at 30 °C.

DNA constructions of bait::reporter fusions were made as follow: the *ALMT1* promoter (1.7 kb and sub-fragments) was PCR-amplified (DNA primers in Supplementary Table 1) and the amplicons inserted in front of the lacZ reporter gene by homologous recombination within the *XmaI*-linearized pSH18–34 vector (Invitrogen), using the yeast strain YPH501. To make the effector construct, the *E. coli B42* activation domain was fused by homologous recombination to the PCR-amplified (Supplementary Table 1) *AtSTOP1* C-terminal coding sequence into the pB42AD (Clontech) vector; expression of this AtSTOP1-B42 fusion was driven under the pGAL inducible promoter. All constructs were checked by DNA sequencing. The primers used for plasmid construction are shown in Supplementary Table 1.

Yeast strain YPH499 transformed with both the bait::reporter and effector constructs were used to quantify DNA-binding activity of STOP1; β-Galactosidase activity assays were performed as described[75]. For all *pALMT1* fragments tested, there was two strains (each with two independent clones): a test strain containing the *pALMT1* vector and the *STOP1::B42* construct and a control strain with the same *pALMT1* vector and the *B42* construct. The β-galactosidase relative activity was calculated as follow: the mean fluorescence of 4-methylumbelliferone (360 nm) per $OD_{600nm}$ measured from the two independent clones (containing the *pALMT1* fragment + *STOP1::B42*) was divided by the mean fluorescence of the corresponding control strain (containing the *pALMT1* fragment + *B42*).

**GFP imaging.** To stain root CWs, seedlings were dipped 5 min in 30 μM PI solution at room temperature and rinsed twice with $H_2O$. Images were collected on a Zeiss LSM780 confocal microscope (Carl Zeiss, France) using a Plan Apochromat × 10/0.45 objective and a Plan-Apochromat × 20/0.8 objective. GFP and PI were excited sequentially with the blue argon ion laser (488 nm) and the DPSS laser (561 nm), respectively. Emitted light was collected from 491 to 535 nm for GFP, and from 600 to 675 nm for PI, using the MBS 488/561 filter. Images were processed using Zen black software (Zeiss).

**Quantification of GFP fluorescence.** To quantify GFP fluorescence in nuclei, images were obtained with a LSM780 confocal microscope (Carl Zeiss, France) using a × 20 dry objective (Plan Apo NA 0.80). GFP was excited with the Argon laser (488 nm), and emitted light was collected between 493 and 538 nm with a pinhole setting of 1.51 a.u.

All nuclei were imaged using the same conditions of gain, offset and resolution (with zoom set to 3), at a constant distance (350 μm) from the root tip. A Z-stack of 15 images (separated by 1 μm distance) imaged representative nuclei on the surface of the root. Images were acquired in 16 bits using Zen black software (Zen black 2012 SP2 Version 11.0), then converted to a maximal projection image. Average nuclei fluorescence intensities were quantified using the Zen blue software (Zen 2 blue edition, version 2.0.0.0), by drawing identical ROI (region of interest) over 3–10 visible (in focus) nuclei for each maximal projection image.

**PERLs-DAB staining.** PERLs-DAB staining was performed as described[12] with minor modifications. DAB intensification was performed for 5 min with a final concentration of 0.00625% DAB. Clearing solution was modified to a higher concentration (8:2:1 hydro-chlorate:water:glycerol (w/v)). Roots were observed and documented with the Axiozoom-V16 macroscope (Zeiss) using the × 2.3 objective.

**GUS histochemical staining.** For *Arabidopsis* seedlings, the GUS staining was conducted as previously described[76]. Briefly, the samples were submersed in a GUS staining solution (50 mM sodium phosphate buffer pH 7.0, 0.01% Triton X-100, 1 mM $K_3Fe(CN)_6$, 1 mM $K_4Fe(CN)_6$, 1 mg.ml$^{-1}$ 5-bromo-4-chloro-3-indolyl β-D-glucuronide) and incubated 1 h at 37 °C. Plants were observed and documented with a DM6000B upright microscope (Leica; × 10 objective) after clearing with chloral hydrate solution as described[12].

For histological sections samples were stained for 20–25 min at 37 °C. Samples were fixed (1% glutaraldehyde, 4% paraformaldehyde in 50 mM phosphate buffer, pH 7.0) for 24 h, followed by stepwise dehydration in ethanol. Embedding was accomplished with Technovit 7100 (Kulzer, Germany). Sections (10 μm) were prepared and stained with ruthenium red. Sections were observed and documented with the axio Zoom.V16 (Zeiss) using the × 2.3 objective.

**Visualization of peroxidase activity.** 4-Chloro-1-naphtol staining was used to visualize peroxidase activity in roots. A volume of 1 μl of a 3% 4-chloro-1-naphthol stock solution (30 mg ml$^{-1}$ in absolute ethanol) and 1 μl of a 3% $H_2O_2$ stock solution were added to 10 ml of 250 mM Tris buffer pH 7.4 resulting in the final staining solution. Plants were stained for 60 s. Staining solution was exchanged with + Pi or − Pi buffer, depending on the sample, and washed three times. Staining was observed and documented with the Leica 6000D ( × 10 objective) for all samples in the same conditions.

**ROS and callose staining.** ROS and callose staining were accomplished as described[12]. ROS were stained using CM-H2DCFDA (6-chloromethyl-2,7-dichlorodihydrofluorescein diacetate; this compound is oxidized by peroxidases in presence of $H_2O_2$) with minor modifications (for example, the phosphate-buffer was replaced by 20 mM 4-(2-hydroxyethyl)piperazine-1-ethanesulfonic acid (HEPES) buffer, pH 6.0). Images were collected on Zeiss LSM780 confocal microscope (Carl Zeiss, France).

**AFM analysis.** Mounting root samples: AFM data were acquired on the primary root from whole seedlings mounted under the microscope. Seedlings were grown 6 days under + Pi prior to transfer under + Pi or − Pi during 0.5–2 h. Seedlings were then glued on a microscope slide (one seedling per slide) as follow: a layer of silicone (MED1–1,356, NuSil Technology LLC, USA), with the thickness of a coverslip slide (130–170 μm), was spread on the slide; silicone was let to dry out for 1 min prior to laid down the seedling on it. To securely anchor the root, a silicone band covering < 100 μm of the root tip was stretched using a syringe needle. Then, a droplet of liquid medium[68] with 10 μM (− Pi) or 500 μM $KH_2PO_4$ (+ Pi), corresponding to the medium (minus agar) in which the seedlings were transferred before AFM analysis, was deposited to cover entirely the seedling to prevent drying. The mounted seedling was then positioned under the AFM instrument. Three seedlings were used per line and per condition.

Data acquisition and apparent Young's modulus cartography: Force–distance curves were obtained with an AFM Dimension 3100 (Bruker, Santa Barbara, USA) with a nanoscope five controller running the Nanoscope 7.3 software. Triangular pyrex nitride cantilever with pyramidal tips of nominal radius of 15 nm, a half-opening angle of 35°, and a nominal spring constant $k = 0.06$ N m$^{-1}$ were used (PNP-TR, NanoWorld AG, Neuchatel, CH). Calibration of photodiode sensitivity was performed using approach-retract curves in liquid on a glass substrate; an average value of 55 nm V$^{-1}$ was commonly obtained. Cantilever spring constant was determined using the thermal tune module of nanoscope and an average value of 0.04 N m$^{-1}$ was commonly found. Positioning of AFM cantilever on plant roots was performed using the Dimension 3100 optical camera so that the targeted zone was at ∼ 500 μm from the end root, as measured with the crosshair (that is, after the optically dense zone corresponding to the meristem). Indentation of plant root surface was performed with approach-retract curves using the picoforce module of nanoscope with a ramp size of 3–4 μm, a scan rate of 1 Hz, and 2,000 points per curve. Ramp size was adjusted using a relative set-point threshold so that an indentation of about 0.5 μm was achieved. Each measurement includes an area of 10 × 40 μm$^2$ using nine zones (5 μm in the direction of the width and 10 μm in the direction of the root axis) into which about 10 force curves were recorded. Thus, a total of about 90 curves were collected per area. Force curves were translated in ASCII text files using YieldFinder[77] to fit the file format of software FORCE v1.1.0 (ref. 78). Indentation curves were fitted according to the author's instructions by a model of Sneddon[79]. The two adjustable parameters, contact point and indentation depth, were manually adjusted to optimize the fitting curve. The cell stiffness as defined by the cell elastic modulus ($E_{cell}$) was obtained according to the formula:

$$F = \frac{2 \times \tan(\alpha) \times E_{cell}}{\pi \times (1 - \mu^2)} \times \Delta z^2$$

where $F$ is the applied force in nN, $\Delta z$ is the indentation in nm, $\alpha$ is the half-opening angle of the tip and $\mu$ is the Poisson's coefficient for typical cell, which is taken as 0.5. Thus, $E_{cell}$ is obtained using the following rearrangement:

$$E_{cell} = \frac{\pi \times (1 - \mu^2)}{2 \times \tan(\alpha)} \times a$$

where $a = \frac{F}{\Delta z^2}$ (obtained from the fitting curves). In this form, $a$ has a unit of $10^6$ kPa.

**Statistics and reproducibility of experiments.** Statistical analyses were performed using Prism 6.0 (GraphPad). Sample size was not predetermined using statistical methods, but took into account the variability of the traits measured, assessed by the s.d.; no samples were excluded from the analyses. Data sets with normal distributions were analysed using one-way or two-way ANOVA, or using two-sided, unpaired Student's t-tests and presented as the means ± s.d. except for Figs 1e and 2b,c and Supplementary Fig. 3a where they are presented as the means ± s.e.m. Data sets with non-normal distribution (for example, AFM measurements) were analysed with Mann–Whitney tests and presented as the median ± interquartile. Data sets with dichotomous distributions (Supplementary Fig. 1d) were analysed with a binomial test. Tukey whiskers extend to the extreme values included in the interval calculated as ± 1.5 IQR (interquartile range), where the IQR is calculated as the third quartile minus the first quartile. The results for statistical significance tests are included in the legend of each figure; n values represent the number of independent experiments performed or the number of seedlings, cells or meristems per condition. The samples were not randomized and the investigators were not blinded to allocation during experiments and outcome assessment. All micrographs are representative of at least two independent experiments.

**Data availability.** All data supporting the findings of this study are available from the corresponding author on request.

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

## Acknowledgements

We thank B. Alonso (CEA, cadarache) for preparing samples in AFM experiments and J. Mutterer (IBMP, Strasbourg) for Image-J M macro design; P. Milani (ENS, Lyon) and A. Peaucelle (INRA, Versailles) for advices in mounting samples in AFM experiments; J.-L. Montillet and D. Rumeau (CEA, cadarache), S. Mari and C. Curie (INRA, Montpellier) and C. Dunand (Univ. Paul Sabatier, Castanet-Tolosan) for fruitful discussions; J. Vermeer (IPB, Zürich) for providing the clones containing the promoters for pSCR, pCo2, pPET and pCASP1 constructs; J. Paz-Ares (CSIC, Madrid) for the *phr1;phl1* double mutant; J. Liu and L. Kochian (Cornell Univ., USA) for the *almt1<sup>KO</sup>;mate<sup>KO</sup>* double mutant carrying the *pALMT1::MATE* construct; P. Benfey (Duke Univ., USA) for the *upb1-1* & *35S::YFP-UP1* lines; G. Desnos for rapeseed seeds; the Nottingham Arabidopsis Stock Centre for providing Arabidopsis KO mutants; the Groupe de Recherches Appliquées en Phytotechnologie (GRAP, cadarache) for plant care. Support for the microscopy equipments was provided by the Région Provence Alpes Côte d'Azur, the Conseil Général des Bouches du Rhône, the French Ministry of Research, the European Union (European Regional Development Fund), the HélioBiotec platform, the CEA and the CNRS; the qRT-PCR machine was funded by Héliobiotech. This work was funded by CEA (APTTOX021401, APTTOX021403), Agence Nationale de la Recherche (ANR-09-BLAN-0118; ANR-12-ADAP-0019) and Investissements d'avenir (DEMETERRES). C.Ba. was supported by Agence Nationale de la Recherche (ANR-12-ADAP-0019); T.Da. was supported by Aix-Marseille-Université, CEA (APTTOX021401) and Investissements d'avenir (DEMETERRES); C.M. was supported by Investissements d'avenir (DEMETERRES); E.L. and E.D. were supported by Agence Nationale de la Recherche (ANR-09-BLAN-0118); M.B. was supported by CEA (APTTOX021401); B.P. was supported by Agence Nationale de la Recherche (ANR Retour Post-doc EmPhos PDOC00301), EMBO Long-Term Fellowship and European Reintegration Grant under the 7th Framework Program of the European Commission (ERG-2010-276662).

## Author contributions

All experimenters contributed to the design of individual experiments and analysis of results, and participated to parts of manuscript preparation. C.Ba. designed and carried out short-term kinetic experiments, contributed to callose and iron-staining experiments, cell length measurements, ICP-MS and AFM experiments. T.Da. prepared plasmids and transgenic plants, performed qRT–PCR and contributed to confocal microscopy. C.G. designed and performed AFM experiments, designed and performed the malate complementation experiment, designed and supervised yeast experiments and carried out the pH experiment with GUS staining. E.L. characterized the EMS mutants and identified most almt1 and stop1 mutations, prepared plasmids and performed the yeast experiments. C.M. designed and performed some GUS staining, the ROS and peroxidase histology and meristem length experiments, participated to confocal microscopy. J.-M.T. designed and performed AFM experiments. A.C. performed mapping of *almt1*[32] and screened EMS mutants for *lpr1* alleles. M.B. made plasmids and transgenic plants, and assisted the yeast experiments and contributed to ICP-MS experiment. C.Br. carried out iron staining experiments, contributed to measurements of roots and cell lengths and genotyping, participated to confocal microscopy. A.H. performed the ICP-MS analysis. J.M. contributed to iron and callose staining experiments and malate complementation experiments. S.C. assisted with GUS histology and qRT-PCR. H.J. assisted and supervised confocal microscopy imaging and quantification. N.L.-B. contributed to genotyping. P.D. contributed to genotyping. B.P. provided materials and tips related to this study. E.D. assisted with qRT-PCR and the design of HRM markers. M.-C.T. assisted with confocal microscopy. J.A. contributed to funding. S.A. helped with discussions about hypotheses, manuscript writing and provided protocols. J.-L.P. supervised, designed and analysed AFM experiments. L.N. contributed for funding, lab administration and to Supplementary Fig. 1a. T.De. gained funding, conceived and supervised the entire project, participated to the design and analysis of all experiments, performed the mutagenesis, mutants screening, genetics analysis and mapping, identified *almt1*[32] and *Stop1*[48], performed experiments on drugs, rapeseed, root length and GUS staining, generated all the figures and tables; wrote the manuscript with feedback from S.A., C.G., H.J., C.M., C.Ba., T.Da., J.-L.P., B.P., L.N., M.-C.T. and E.D.

## Additional information

**Competing interests:** The authors declare no competing financial interests.

