## [Peer Review File · Nature Communications]

Reviewers' comments:

Reviewer #1 (Remarks to the Author):

Balergue et al present an impressively data-rich manuscript resulting from clearly a large undertaking by a number of researchers (with 5 first authors). Their study concludes there are two responses to P deficiency that reduce root growth. Firstly, they conclude there is a growth inhibition that occurs within the first two hours that is dependent upon the transcription factor STOP1 and the malate transporter ALMT1, which immobilizes Fe stimulating LPR1 peroxidase activity to cause wall stiffening. Secondly they propose a reduction in cell proliferation at the meristem after a few days, this is proposed to be LPR1 dependent but not involving STOP1-ALMT1. This is an interesting manuscript that deserves a good hearing. Whilst generally an excellent piece of work, there remains several questions in regard to interpretation, and there needs to be an improvement in the presentation of several key bits of information.

Major points

1. It is very important that the authors state the pH at which these experiments performed—according to the reference in the methods most were at pH5.8. Is that correct? This is also likely due to the root phenotypes of the stop1 ko – i.e. they are not shorter than WT +P.

The reason for stating pH is several fold. STOP1 is induced by low pH. Can the authors show whether this activation at low pH of STOP1 is due to low P or to a pH effect, or are there multiple reasons that STOP1 activates?

Secondly, and significantly at low pH or high P, P deficiency is more common. Can the authors address the significance of P deficiency response at pH 5.8? Does this response also occur at low and high pH, when P deficiency is more common? This impacts majorly on the significance of this work.

2. There appears to be a tight link between Fe and ALMT1, which is not fully explored or discussed by the authors. WRKY46 is a transcription factor activated by Fe deficiency (Ying Yan et al 2016 Plant Phys) and is a transcriptional repressor of ALMT1 (Ding et al., 2013 Plant Journal). Fe staining in the root appears to require STOP1 transcriptional activation of ALMT1. Therefore you have a situation of relative high Fe (without P) stimulating ALMT1 and low Fe repressing ALMT1. This should be discussed. Furthermore, it would instructive if the authors made a specific statement as to how they avoided the Fe toxicity effect of low P, and to ensure that what they are observing is not a Fe toxicity response. P9 L258 states that the effect of low P on root growth is abolished at low Fe or at higher pH, which the authors explain is due to the lack of peroxidase activity – but is ALMT1 even induced in such conditions? Is this response of ALMT really to low P or to greater Fe availability? This impacts majorly on the interpretation of the authors work.

3. The genetic evidence appears tight but in terms of transport there is an issue. ALMT1 has to be activated, presence of the transcript alone does not guarantee that any protein

produced is actively exuding malate. Malate currents are very small in ALMT without AI present (Hoekenga et al 2005, PNAS) – or potentially a transactivating anion (see Ramesh et al 2015 Nature Comms). There is a possibility that Fe may activate ALMT1, or some unknown factor. This needs to be acknowledged in the manuscript. This impacts majorly on mechanism underlying the authors findings.

4. Whilst most of the data is solid and quantitative in this manuscript, Fig 1f and g, Fig3a, Fig4j, 4k, Sup2h and i, Supp 3-7 are all representative plants following a staining procedure or of a fluorescent protein. In particular, fig 3a is lynchpin data. To show one representative image and no quantification is not acceptable. Fig1g is equally poor. In no way do I doubt the authors honesty and integrity but these data need to be quantified, or at least shown how representative they are by use of the supplementary figures. I see this as necessary to ensure the integrity of the work and to maintain the standards of the journal.

5. The non-use of the equivalent WT to the T-DNA KO plants is an issue. The correct control for these plants is not Coler106. This needs to be included.

6. Please indicate/mark/state significance for important figs where not obvious such as 1e, 2b. Furthermore, it is potentially misleading to show 1e as relative expression normalised to +P. It may be neater but what is the actual copy number – how highly is ALMT1 etc expressed – even compared to a control gene would be better than double transformed data – at least in the supplementary data. This is linked to point 3 above.

7. P8L226 if stating there is a correlation please show a plot and regression.

8. Whilst, the manuscript still is largely novel several citations are missing that point to a role of ALMT proteins in iron deficiency such as Liang et al 2013 Plant Phys, furthermore, the references cited above on ALMT function and regulation are important as this manuscript builds upon some of these findings.

9. Generally for a multidisciplinary journal it is good to show that the findings are more applicable than just to Arabidopsis. Is there a simple experiment to show that these findings have any relevance to major crops such as wheat and rice?

Minor comments

The use of hyperbolic terms should be toned down. Use of exquisite, salient, profoundly, remarkably, indeed, etc should be limited.

P3 L33 How much land is affected by P deficiency, what impact does this have on yield?

P3L39 chose

P3L64 which cell types?

P4L73 advise by -P, instead of -P condition, check throughout

P4L76 revise – encoded by a direct STOP1 target gene, as rather than by?

P4L96 Please very briefly explain the data underpinning the assumption it is dominant negative – this is an important piece of information. Also cite sup 1d?

Fig Supp 2f is seemingly random – why is this data included?

P7L171 delete already

In Fig 3a, why is there greater Fe staining in the +P. Does this impact upon the interpretation of the results?

In Fig 5b, there is staining shown in the *lpr1/lpr2* knockout in the meristem – can the authors comment why this is present – and why it doesn't match with the Fe deposition pattern?

Reviewer #2 (Remarks to the Author):

I carefully read the MS, and found that the MS provides new insight of Stop1-ALMT1 system. In addition, this clearer demonstrates pleiotropic function of malate released from the roots, Experiments were carefully conducted, and the predicted model by the observation of the suppressed-growth in the wild type occurred P-deprivation was carefully analyzed in relation to Stop1-ALMT1 system. All the data sounds well to describe the function of malate excretion for regulating root growth, which mediates inactivation of Fe in the apoplast by the formation of malate-Fe. I have no "major criticism" to the overall story, and experimentation. However, I have some questions, and a series of comments that may be considered when the authors finalized the MS.

- 1) I found that the mutant analyses were carefully done. Among the mutation in the *stop1* allele, two showed dominant negative phenotype, while the remaining others were recessive. Could you explain the possible mechanisms of dominant negative?
- 2) I agree with the less contribution of the MATE-dependent citrate excretion in Arabidopsis for regulating the Fe-chelator dependent growth suppression under P-stress. However, in many crops (e.g. tobacco), citrate excretion controlled by STOP1 plays major role instead of ALMT1-dependent malate excretion (at least for Al tolerance). As you may know, STOP1-ALMT1/MATE system is ubiquitous among crops (and, possibly among land-plant species). I would like to recommend the authors to expand the finding-concept with Arabidopsis to other crops, in particular citrate-excreting species through STOP1/MATE system.

Reviewer #3 (Remarks to the Author):

In the present manuscript, Balzergue and colleagues report the identification of a novel mechanism involved in the adjustment of primary root elongation in response to phosphorus (P) supply. According to their results, this mechanism involves malate extrusion and the consequent increase in apoplastic Fe deposition and peroxidase activity in root tips, which they link to the increased cell wall stiffness and the inhibition of cell expansion observed in P-deficient roots.

Previously, other groups have shown that the strong inhibition of primary root elongation under low P is highly dependent on the external Fe concentrations and requires the ferroxidases LPR1 and LPR2 and the P5-type ATPase PDR5. In the present study, the authors set out to identify new molecular players by screening EMS mutants that maintain primary root elongation under -P. Besides isolating new alleles of the previously characterized *lpr1* mutant, the authors also identified several mutant alleles of the malate efflux channel ALMT1 and the transcription factor STOP1, which have been shown in earlier studies to play a role in malate exudation in roots. They also show that whereas STOP1 is stimulated by P deficiency post-translationally, ALMT1 is up-regulated by -P at the transcriptional level and in a STOP1-dependent manner. With the help of atomic force microscopy, the authors demonstrate that cell wall stiffness is quickly increased when WT plants are transferred to -P, and that this effect is not observed in *almt1*, *stop1* and *lpr1lpr2* mutants. The altered cell wall stiffness is then related to the differential cell elongation shown by P-deficient wild-type and mutant plants. As shown by pharmacological experiments with peroxidase inhibitors and one method for in vivo peroxidase staining, local P deprivation appears to enhance peroxidase activity in primary roots and, when the activity of these enzymes is inhibited, the elongation of P-deficient roots can be maintained. Such effect was hypothesized to be related to a peroxidase-dependent inhibition of cell wall expansion.

The manuscript provides novel insights on how plants adjust their root growth in response to severe P limitation. However, there are several open points that still deserve more attention:

- 1) One of the major issues of this study is that the link between malate exudation, peroxidase activity and cell wall stiffening is not satisfactorily worked out. In addition, it seems that large aspects of what is known so far about the effects of P deficiency in root elongation have not been sufficiently integrated in the current manuscript. One sign for this apparent weakness is that the authors do not present a clear hierarchy between peroxidase activity, Fe accumulation and cell wall stiffening in their final model (Fig. 6). For instance, the authors suggest that the differential levels of cell wall stiffness recorded in roots of wild-type plants and the P-insensitive mutants was associated to peroxidase-dependent changes in cell wall composition. However, no evidence for any cell wall modification is presented. Thus, this link still remains largely based on correlative evidence. Furthermore, it has been recently shown that low P supply results in a prominent cell wall modification, i.e. a massive deposition of callose in the stem cell niche and in cortical cells of the elongation zone (Müller et al., 2015; *Dev Cell* 33, 216-230). Surprisingly, in the present study the authors do not show whether callose deposition is altered in roots of the *almt1* and *stop1* mutants or

whether callose deposition could explain the observed changes in cell wall stiffness. Furthermore, according to Müller et al. (2015), increased callose deposition in P-deficient roots is associated with Fe-dependent ROS formation. Therefore, rather than revealing a functional role in cell wall remodeling, the reported differences in peroxidase activity may, alternatively, only reflect the activation of scavenging mechanisms in response to a differential Fe-dependent ROS accumulation in the investigated mutants. This possibility should be tackled by assessing ROS distribution and callose deposition in *almt1* and *stop1* mutants as well as in plants treated with peroxidase inhibitors.

2) I'm not sure if the conclusions derived from the cell type-specific activity of STOP1 (and consequently of ALMT1) are sufficiently justified. For instance, is the restored primary root sensitivity towards -P in *pSCR::STOP1/stop1ko* accompanied by increased Fe accumulation and peroxidase activity in the elongation zone? In addition, according to STOP1-promoter activity, this transcription factor is more strongly expressed in epidermal cells (Suppl Fig. 3a). Thus, why did the authors not try to restore STOP1 expression in this cell layer? Moreover, the title of Fig.5 does not really make sense and should be related to the expression of STOP and not the cell types themselves.

3) In general, the micrographs showing peroxidase activity in roots are of low quality. In most cases, only minor differences between peroxidase activity in roots of wild type and *almt1* and *stop1* mutants can be seen. As these results are of key relevance in the study, I would recommend improving the overall quality of these micrographs. One alternative would be to enlarge the root pictures shown in Fig. 4j and k and/or take pictures of stained roots at higher magnifications. Perhaps some of the mutant alleles could be moved to a Suppl. Figure to increase space in Fig. 4. In addition, in order to establish whether the increased peroxidase activity precedes the inhibition of cell elongation, the enzymatic activity should also be assessed before cell elongation of Col-0 roots is severely arrested by P deficiency. In this regard, including an alternative approach to further demonstrate differences in peroxidase activity would also help to reinforce these datasets.

4) The effects of malate, which plays a central role in the proposed model, have been poorly investigated leading to the following open questions: Are the changes in primary root length in response to exogenous malate supply statistically significant (Fig. 1d), and does external malate supply lead to a differential response of the EZ and RAM incl. corresponding Fe accumulation? Can the external supply of malate increase apoplastic Fe accumulation and peroxidase activity in the elongation zone of *almt1* and *stop1* mutants?

Reviewer #4 (Remarks to the Author):

The authors have demonstrated the link between ALMT1 and STOP and Pi starvation. These two proteins have been already described then, the authors should pay more attention to the references included in the introduction regarding these two proteins (e.g.: STOP1 (<https://www.ncbi.nlm.nih.gov/pmc/articles/PMC2675709/>) and ALMT1 (<https://www.ncbi.nlm.nih.gov/pubmed/23624855/>)). A recent paper linking both proteins

could be included (<https://www.ncbi.nlm.nih.gov/pubmed/25627216/>).

The figures are very complete but part of them are not used or described in the manuscript or sometime described very briefly late in the discussion. To illustrate this situation, the description of UPB1 appears in the last line of the discussion. To my opinion, UPB1 and the ROS homeostasis regulation could be included early in the paper and should be more discussed (<https://www.ncbi.nlm.nih.gov/pubmed/21074051/>).

The authors need to clarify the choice of the EMS mutants and justify the interest to present a large battery of *almt1* and *stop1* EMS mutants together with the two corresponding T-DNA mutant. The phenotypes are more or less similar then the use of one mutant is sufficient.

The phenotypic variation between the mutants is interesting and could be deepened regarding the expression of *almt1* and *stop1* in the different mutants

The lack of statistical analysis need to be corrected.

I am perhaps wrong, but the discussion ends with the idea of ROS homeostasis and peroxidases necessary for low phosphate root growth. Which is fine for me, but which is different from the initial message.

New and very interesting results are presented by the paper of Desnos and collaborators. But the quality and the readability of he paper will be largely improved with the reformatting of the figures and a better state of the art regarding ALMT1 and STOP1

Reviewer #5 (Remarks to the Author):

As per the editorial request, I am only commenting on the part of this paper in which the author's use AFM to characterize plant cell wall stiffening. I note 3 major concerns (points 3,5, 6 and 7 below).

I commend the authors for their method to secure the roots to the cover slip (and their clear description) since securely held samples is an important part of AFM methodology and is difficult to achieve.

1. However with living root samples it is also important to check that temporal changes did not occur purely due to the root drying out./ being affected by chemical interaction with the adhesive. Do the authors have data at e.g. 2 hrs in +Pi to indicate that stiffening was not occurring purely due to temporal changes in root morphology due to these sort of concerns? (e.g. +Pi data at 1 and 2h to compare with -Pi data in Fig 4a)

They state (line 698) that an average value of spring constant was 0.04 N/m.

2. To gain accurate force values it is necessary to use the precise spring constant of the AFM tip at the point of use. I think they may have done this since, from my reading of their reference 74, it is a part of the Yieldfinder software but it would be good to state this.

Their method in which they adjusted the ramp size to give indentations of approximately the same depth for all samples is also to be commended since comparing indentations of similar size is an important step when comparing indentation data from samples of different stiffness.

3. A major question re the methodology in this paper is the location of targeted zone at ~ 500 μm from the "end root" (line 700) for all roots. If roots are a different size either through sample to sample variability of "identical roots", or because root growth has been inhibited, is it correct that the transition zone in all the roots will be the same distance from the end of the root? The whole stiffening argument hangs on being sure that you are indenting like zone with like zone.

4. Lines 711-713 mentions that the fit was manually adjusted to optimise the fitting curve. Please could the authors provide some data to describe the goodness of fit they typically get. E.g. Ref 75 Fig 3 shows data points and a line of fit obtained. (I note in passing that this reference describes the formulation of the FORCE software for animal cells with measured stiffnesses in the range of 1-10kPa, one or two orders of magnitude compared to the stiffnesses being reported in this paper. The authors could consider commenting on this.

5. My second major point is that Line 224 claims that a higher stiffness is not significantly detected for *top1*, *alm1* and *lpr1:lpr2* lines. However the authors fail to comment that the *lpr1:lpr2* data presented actually indicates a significant softening of the cell wall in $-\text{Pi}$.

6. Line 225-226 makes the same "not higher stiffness" claim for WT grown in $-\text{P}$ and $-\text{Pi-Fe}$ but again fails to comment that that data present shows a significant softening.

7. My final major comment is that the authors, correctly, state that they are measuring transverse stiffness (line 216). However my understanding is that cell growth occurs mainly in the axial direction. In line 228 the paper suggests a simple causal scenario where $-\text{Pi}$ triggers the CW stiffening of pre-elongated cells to restrict their elongation but fails to comment on the large assumption that axial stiffness will be affected in the same way as transverse stiffness.

Answers (in blue) to Reviewers' comments:

Reviewer #1 (Remarks to the Author):

Balergue et al present an impressively data-rich manuscript resulting from clearly a large undertaking by a number of researchers (with 5 first authors). Their study concludes there are two responses to P deficiency that reduce root growth. Firstly, they conclude there is a growth inhibition that occurs within the first two hours that is dependent upon the transcription factor STOP1 and the malate transporter ALMT1, which immobilizes Fe stimulating LPR1 peroxidase activity to cause wall stiffening. Secondly they propose a reduction in cell proliferation at the meristem after a few days, this is proposed to be LPR1 dependent but not involving STOP1-ALMT1. This is an interesting manuscript that deserves a good hearing. Whilst generally an excellent piece of work, there remains several questions in regard to interpretation, and there needs to be an improvement in the presentation of several key bits of information.

Major points

1. It is very important that the authors state the pH at which these experiments performed– according to the reference in the methods most were at pH5.8. Is that correct? This is also likely due to the root phenotypes of the *stop1* ko – i.e. they are not shorter than WT +P.

Our *stop1* mutants (as well as *almt1* and *lpr1*) were isolated at pH5.6 (as in Svistoonoff et al., 2007) and since more than ten years we are using a growth medium at pH5.6-5.8 (as indicated in Reymond et al., 2006). This is now explicitly indicated in the Methods (P18).

The reason for stating pH is several fold. STOP1 is induced by low pH. Can the authors show whether this activation at low pH of STOP1 is due to low P or to a pH effect, or are there multiple reasons that STOP1 activates?

Since the +Pi and the –Pi media are at the same pH, the effect we observe in phenotypes and gene expressions are due to the Pi status of the media. In addition, we performed a new experiment where we have grown at pH4.4, 5.8 or 7.1 the WT line carrying the *pALMT1::GUS* reporter. Results confirm that the transcription of *ALMT1* (and therefore the activation of STOP1) is stronger at pH4.4, than at pH5.8, and is abolished at pH7.1 (see P7 L211 and Supplementary Fig. 5a).

All these results indicate that STOP1 activity is also sensitive to the Pi status of the environment (P8 L225), in addition to the low-pH and to Al³⁺ as described in the literature.

Secondly, and significantly at low pH or high P, P deficiency is more common. Can the authors address the significance of P deficiency response at pH 5.8? Does this response also occur at low and high pH, when P deficiency is more common? This impacts majorly on the significance of this work.

We previously showed (Supplemental Fig.1 in Svistoonoff et al., 2007) that the primary root growth of *Arabidopsis* is inhibited by the –Pi medium at acidic pH (from pH5.3 to pH6.3): the more acidic, the stronger inhibition; above pH6.3 there is no primary root growth inhibition. This lack of inhibition under higher pH makes sense knowing that the root growth inhibition we describe in this work is Fe-dependent, and that at neutral and basic pH iron ions precipitate. In addition, under higher pH, *ALMT1* is not transcribed (see the new result in Supplementary Fig. 5a). This growth inhibition by –Pi seems therefore specific of acidic conditions.

See also our comments below (point 2, “P9L258”).

2. There appears to be a tight link between Fe and ALMT1, which is not fully explored or discussed by the authors. WRKY46 is a transcription factor activated by Fe deficiency (Ying Yan et al 2016 Plant Phys) and is a transcriptional repressor of ALMT1 (Ding et al., 2013 Plant Journal). Fe staining in the root appears to require STOP1 transcriptional activation of ALMT1. Therefore you have a situation of relative high Fe (without P) stimulating ALMT1 and low Fe repressing ALMT1. This should be discussed.

In a new experiment, we have grown the homozygous *wrky46*^{KO} mutant in –Pi. The result shows that it behaves like the WT (P8 216 and Supplementary Fig. 6), indicating that WRKY46 is not crucial for modulating the root growth arrest in –Pi.

Furthermore, it would be instructive if the authors made a specific statement as to how they avoided the Fe toxicity effect of low P, and to ensure that what they are observing is not a Fe toxicity response.

The toxicity of Fe in $-Pi$ condition is a concept developed by Ward and coll. (Plant Physiology 2008, 147: 1181). Unfortunately, these authors did not show or demonstrate any toxicity symptoms in their article; therefore we cannot assess this “toxicity”. This is why we prefer, instead of “toxicity”, the concept of balance or ratio of Fe over Pi that is more appropriate to encapsulate this phenomenon: at higher Pi levels, the Fe is complexed with Pi anions and cannot activate the root growth arrest, whereas in $-Pi$, the “free” Fe (not complexed by $-Pi$) can activate the growth arrest through ROS production (Müller et al. 2015); ROS can be both signalling molecules and the substrate of peroxidases.

P9 L258 states that the effect of low P on root growth is abolished at low Fe or at higher pH, which the authors explain is due to the lack of peroxidase activity – but is ALMT1 even induced in such conditions? Is this response of ALMT1 really to low P or to greater Fe availability? This impacts majorly on the interpretation of the authors work.

To answer this key point, we performed a new experiment: we have grown the *pALMT1::GUS* line in $-Pi$ with or without Fe (Supplementary Fig. 5a). The result shows that omitting iron in the growth media does not impair the stimulation of *ALMT1* expression by $-Pi$ (compare $-Pi-Fe$ with $-Pi+Fe$ in Supplementary Fig. 5a). This new result supports the view that iron does not have a crucial role in the stimulation of *ALMT1* expression in $-Pi$; instead, it is the low-Pi status of the medium that is important (see text Page 8).

By contrast, as shown in Supplementary Fig. 5a (see above), at pH7.1 *ALMT1::GUS* is not expressed in + and $-Pi$ (most probably because STOP1 is not activated). This could be part of the explanation of the lack of growth inhibition in $-Pi$ at this pH. However, even when *ALMT1* is constitutively expressed, there is no growth inhibition (Supplementary Fig. 5b). One explanation could be that at higher pH the Fe precipitates and therefore cannot inhibit root growth, although seedlings are in $-Pi$ condition. This does not exclude, as suggested by reviewer (see below at point 3), that $Fe^{2+/3+}$ could be an activator of ALMT1 protein at acidic pH, as Al^{3+} is. However, by contrast to the wheat TaALMT1 for example, there are conflicting results about the Al^{3+} -activation of malate efflux by Arabidopsis ALMT1 (Sasaki et al., Biochimica et Biophysica Acta 1858 (2016) 1427–1435 ; Sasaki et al., Plant Cell Physiol. 55(12): 2126–2138; 2014). We commented this point in the revised text (Discussion Page 15).

3. The genetic evidence appears tight but in terms of transport there is an issue. ALMT1 has to be activated, presence of the transcript alone does not guarantee that any protein produced is actively exuding malate. Malate currents are very small in ALMT without Al present (Hoekenga et al 2005, PNAS) – or potentially a transactivating anion (see Ramesh et al 2015 Nature Comms). There is a possibility that Fe may activate ALMT1, or some unknown factor. This needs to be acknowledged in the manuscript. This impacts majorly on mechanism underlying the authors findings.

We commented on this key point in the revised text (Discussion Page 15).

4. Whilst most of the data is solid and quantitative in this manuscript, Fig 1f and g, Fig3a, Fig4j, 4k, Sup2h and i, Supp 3-7 are all representative plants following a staining procedure or of a fluorescent protein. In particular, fig 3a is lynchpin data. To show one representative image and no quantification is not acceptable. Fig1g is equally poor. In no way do I doubt the authors honesty and integrity but these data need to be quantified, or at least shown how representative they are by use of the supplementary figures. I see this as necessary to ensure the integrity of the work and to maintain the standards of the journal.

Fig1f: More pictures are shown in the Supplementary Fig.16a. New pictures (although not on cross sections) of a similar experiment fully support this result (Sup.Fig5a).

Fig1g: We have repeated this experiment and quantified the GFP-fluorescence in nuclei (Fig.1g and Supplementary Fig.7c).

Fig3a: Fig3a and SupFig8b are already from two independent experiments. Anyway, more pictures of these two experiments are shown in the Supplementary Fig.17.

Fig4j,k (now **Fig.5g,h**): more pictures are shown in the Supplementary Fig.18a,b.

SupFig.2h (now **SupFig.4b**): We have repeated this experiment and made a figure (Supplementary Fig.15)

SupFig.2i (now **SupFig.4c**): This figure already contained several seedlings for each condition. Anyway, we have repeated this experiment and made a figure (Supplementary Fig.15). In addition, similar results (from different experiments) about *pALMT1::GUS* expression in WT and in a *stop1* background are shown in SupFig.12a,b and SupFig.13b (and in their corresponding newly replicated experiments shown in SupFig.20 and SupFig.21).

SupFig. 3a (now **SupFig.7a**): More pictures are shown in the Supplementary Fig.16b.

SupFig. 3c (now **SupFig.7c**): We have repeated this experiment and quantified the GFP-fluorescence in nuclei.

SupFig. 4b (now **SupFig.8b**): more pictures are shown in the Supplementary Fig.17b.

SupFig. 5b,c (now **SupFig.10c,b**): more pictures are shown in the Supplementary Fig.18c,d.

SupFig. 6a,b (now **SupFig. 12a,b**): we have repeated this experiment and made a figure (Supplementary Fig. 20).

SupFig.7b (now **SupFig.13b**): we have repeated this experiment and made a figure (Supplementary Fig.21).

5. The non-use of the equivalent WT to the T-DNA KO plants is an issue. The correct control for these plants is not Col-0. This needs to be included.

Col-0 is the parental line of both the T-DNA and Col^{er105} lines.

Here are the primary root lengths of five-day old seedlings grown in + or -Pi (two independent experiments):

Experiment n°1: +Pi: Col-0 = 22.7 +/-2.8 mm, Col^{er105} = 24.7 +/-2.4mm; -Pi: Col-0 = 8.3 +/-1.6 mm; Col^{er105} = 7.4 +/-1.4mm (mean +/- s.d., n = 14-20).

Experiment n°2: +Pi: Col-0 = 18 +/-2.1 mm, Col^{er105} = 20 +/-1.7mm; -Pi: Col-0 = 6.9 +/-1.7 mm; Col^{er105} = 7.3 +/-1.6mm (mean +/- s.d., n = 32-38).

These results show that WT Col^{er105} and WT Col-0 seedlings do not behave differently, in + or -Pi.

We think it is not essential to include this result in the article.

6. Please indicate/mark/state significance for important figs where not obvious such as 1e, 2b. Furthermore, it is potentially misleading to show 1e as relative expression normalised to +P. It may be neater but what is the actual copy number – how highly is ALMT1 etc expressed – even compared to a control gene would be better than double transformed data – at least in the supplementary data. This is linked to point 3 above.

Statistics and normalization in Fig.1e and SupFig.2b : expression levels (not normalized) with statistics are given in Supplementary Fig.14a,b.

Statistics in Fig.2b,c: they are now added to the figure.

7. P8L226 if stating there is a correlation please show a plot and regression.

We substituted this occurrence of the word “correlation” by “reciprocal relation” (Page 11).

8. Whilst, the manuscript still is largely novel several citations are missing that point to a role of ALMT proteins in iron deficiency such as Liang et al 2013 Plant Phys, furthermore, the references cited above on ALMT function and regulation are important as this manuscript builds upon some of these findings.

The article of Liang et al. (Plant Phys, 2013) is indeed interesting because it shows that, in soybean roots, *GmALMT1* expression and malate exudation are regulated by Al³⁺, pH and also by -Pi. These authors found that, at the very acidic pH4.3 (without Al³⁺), -Pi increases the expression of *GmALMT1*. However, there is a difference with Arabidopsis: *GmALMT1* is three times less expressed at pH4.3 (+Pi, without Al³⁺) than at pH5.8, whereas Arabidopsis *ALMT1* is more expressed at very acidic pH4.4 than at pH5.8 (see Supplementary Fig. 5a). Anyway, we cited this article (P14 L439) as well as several articles about the ALMT1 function and regulation that were missing in the first version.

9. Generally for a multidisciplinary journal it is good to show that the findings are more applicable than just to Arabidopsis. Is there a simple experiment to show that these findings have any relevance to major crops such as wheat and rice?

We have added a new result showing that the primary root of seedlings from the rapeseed crop (*Brassica napus*) is also inhibited in -Pi (Supplementary Fig. 3).

Minor comments

The use of hyperbolic terms should be toned down. Use of exquisite, salient, profoundly, remarkably, indeed, etc should be limited.

We removed some but we also consider that, since *Nature Communications* is a multidisciplinary journal, it is important to highlight (for those outside the field) which results are considered as critical or noteworthy.

P3 L33 How much land is affected by P deficiency, what impact does this have on yield?

We change the sentence (now P4 L82).

P3L39 chose

Corrected (now P4 L87).

P3L64 which cell types?

Corrected (now P4 L112).

P4L73 advise by -P, instead of -P condition, check throughout

Corrected.

P4L76 revise – encoded by a direct STOP1 target gene, as rather than by?

Sentence modified (P5 L124).

P4L96 Please very briefly explain the data underpinning the assumption it is dominant negative – this is an important piece of information. Also cite sup 1d?

Done (P5 L145).

Fig Supp 2f is seemingly random – why is this data included?

Sup Fig. 2f (now SupFig.2g) is mentioned page 7 line 194; this result is important to demonstrate that *pALMT1* is a direct target of STOP1 and it confirms a result recently published by others using another technique (Tokizawa et al. 2015).

P7L171 delete already

Done.

In Fig 3a, why is there greater Fe staining in the +P. Does this impact upon the interpretation of the results?

The greater Fe staining in +Pi was already observed and discussed by Müller et al. (2015). This observation does not undermine the interpretation of the results because —according to our model— in +Pi, Fe is complexed with Pi and therefore cannot inhibit root growth (See model in Fig.7).

In Fig 5b, there is staining shown in the *lpr1/lpr2* knockout in the meristem – can the authors comment why this is present – and why it doesn't match with the Fe deposition pattern?

In Sup Fig.5b (now SupFig.10c) the staining seen at the root tip is in the columella, not in the meristem. Since there are many peroxidases expressed in arabidopsis roots, it is possible that this staining comes from the activity of peroxidase(s) not related with the mechanism studied in this work.

Reviewer #2 (Remarks to the Author):

I carefully read the MS, and found that the MS provides new insight of Stop1-ALMT1 system. In addition, this clearer demonstrates pleiotropic function of malate released from the roots, Experiments were carefully conducted, and the predicted model by the observation of the suppressed-growth in the wild type occurred P-deprivation was carefully analyzed in relation to Stop1-ALMT1 system. All the data sounds well to describe the function of malate excretion for regulating root growth, which mediates inactivation of Fe in the apoplast by the formation of malate-Fe. I have no “major criticism” to the overall story, and experimentation. However, I have some questions, and a series of comments that may be considered when the authors finalized the MS.

1) I found that the mutant analyses were carefully done. Among the mutation in the stop1 allele, two showed dominant negative phenotype, while the remaining others were recessive. Could you explain the possible mechanisms of dominant negative?

We have added a sentence explaining the possible molecular mechanisms for the dominant negative behavior (P6 L146).

2) I agree with the less contribution of the MATE-dependent citrate excretion in Arabidopsis for regulating the Fe-chelator dependent growth suppression under P-stress. However, in many crops (e.g. tobacco), citrate excretion controlled by STOP1 plays major role instead of ALMT1-dependent malate excretion (at least for Al tolerance). As you may know, STOP1-ALMT1/MATE system is ubiquitous among crops (and, possibly among land-plant species). I would like to recommend the authors to expand the finding-concept with Arabidopsis to other crops, in particular citrate-excreting species through STOP1/MATE system.

We added a new result showing that MATE cannot replace ALMT1 when expressed under the control of ALMT1 promoter (P6 L175 and Supplementary Fig.2d). This indicates that citrate exuded at the root tip, in the same tissues where ALMT1 is normally expressed, is not able to replace malate in inhibiting root growth in -Pi. Extending this concept to crop species will depend whether they exude malate. In low-Pi, *Brassica napus* is known to exude malate, as well as citrate (Hoffland et al., Plant and Soil 113, 161-165 (1989)). We have added a new result showing that its primary root is also inhibited in -Pi (P6 L164 and Supplementary Fig.3).

Reviewer #3 (Remarks to the Author):

In the present manuscript, Balzergue and colleagues report the identification of a novel mechanism involved in the adjustment of primary root elongation in response to phosphorus (P) supply. According to their results, this mechanism involves malate extrusion and the consequent increase in apoplastic Fe deposition and peroxidase activity in root tips, which they link to the increased cell wall stiffness and the inhibition of cell expansion observed in P-deficient roots.

Previously, other groups have shown that the strong inhibition of primary root elongation under low P is highly dependent on the external Fe concentrations and requires the ferroxidases LPR1 and LPR2 and the P5-type ATPase PDR5. In the present study, the authors set out to identify new molecular players by screening EMS mutants that maintain primary root elongation under -P. Besides isolating new alleles of the previously characterized *lpr1* mutant, the authors also identified several mutant alleles of the malate efflux channel ALMT1 and the transcription factor STOP1, which have been shown in earlier studies to play a role in malate exudation in roots. They also show that whereas STOP1 is stimulated by P deficiency post-translationally, ALMT1 is up-regulated by -P at the transcriptional level and in a STOP1-dependent manner. With the help of atomic force microscopy, the authors demonstrate that cell wall stiffness is quickly increased when WT plants are transferred to -P, and

that this effect is not observed in *almt1*, *stop1* and *lpr1lpr2* mutants. The altered cell wall stiffness is then related to the differential cell elongation shown by P-deficient wild-type and mutant plants. As shown by pharmacological experiments with peroxidase inhibitors and one method for in vivo peroxidase staining, local P deprivation appears to enhance peroxidase activity in primary roots and, when the activity of these enzymes is inhibited, the elongation of P-deficient roots can be maintained. Such effect was hypothesized to be related to a peroxidase-dependent inhibition of cell wall expansion.

The manuscript provides novel insights on how plants adjust their root growth in response to severe P limitation. However, there are several open points that still deserve more attention:

1) One of the major issues of this study is that the link between malate exudation, peroxidase activity and cell wall stiffening is not satisfactorily worked out. In addition, it seems that large aspects of what is known so far about the effects of P deficiency in root elongation have not been sufficiently integrated in the current manuscript. One sign for this apparent weakness is that the authors do not present a clear hierarchy between peroxidase activity, Fe accumulation and cell wall stiffening in their final model (Fig. 6). For instance, the authors suggest that the differential levels of cell wall stiffness recorded in roots of wild-type plants and the P-insensitive mutants was associated to peroxidase-dependent changes in cell wall composition. However, no evidence for any cell wall modification is presented. Thus, this link still remains largely based on correlative evidence. Furthermore, it has been recently shown that low P supply results in a prominent cell wall modification, i.e. a massive deposition of callose in the stem cell niche and in cortical cells of the elongation

zone (Müller et al., 2015; Dev Cell 33, 216-230). Surprisingly, in the present study the authors do not show whether callose deposition is altered in roots of the *almt1* and *stop1* mutants or whether callose deposition could explain the observed changes in cell wall stiffness. Furthermore, according to Müller et al. (2015), increased callose deposition in P-deficient roots is associated with Fe-dependent ROS formation. Therefore, rather than revealing a functional role in cell wall remodeling, the reported differences in peroxidase activity may, alternatively, only reflect the activation of scavenging mechanisms in response to a differential Fe-dependent ROS accumulation in the investigated mutants. This possibility should be tackled by assessing ROS distribution and callose deposition in *almt1* and *stop1* mutants as well as in plants treated with peroxidase inhibitors.

In our manuscript, we did not suggest that the cell wall stiffness is associated to peroxidase-dependent changes in cell wall composition. However, as mentioned in the introduction of the manuscript, Müller et al. (2015) reported that when grown in -Pi the WT, but not the *lpr1;lpr2* double mutant, accumulates callose in the cell wall of SCN and the elongation zone. We therefore tested whether the *stop1* and *almt1* mutants are altered in this accumulation of callose (P9 L271 and Fig.3b). This new result shows that in -Pi, these mutants still accumulate callose in the SCN but do not in the elongation zone, mirroring their accumulation of iron. This significant result strengthens the idea that the *stop1* and *almt1* mutations uncouple the response in the elongation zone from the response in the RAM. Note that exogenous malate complements the *almt1* mutant for callose (and iron) deposition in the elongation zone (a new result showed in Supplementary Fig. 9 and P9 L274). Does callose deposition is responsible of the increased cell wall stiffness we observed? We prefer to stay prudent about this hypothesis because callose deposition could also be a late, secondary effect of the -Pi stress.

We also observed ROS in the transition and differentiation zone of WT in -Pi (P12 L373 and Supplementary Fig.11); these ROS staining appear less intense than in Müller et al (2015) (in particular in the meristem) probably because our protocol was less sensitive. This new experiment shows less accumulation of ROS in +Pi+Fe, -Pi-Fe and -Pi+Fe+SHAM conditions, compared to -Pi+Fe. All these results are now included and discussed in the revised manuscript.

That the chemical inhibitors (three structurally different drugs) of peroxidase activity suppress cell wall stiffening and fully restore root growth strongly suggests that peroxidase activity (as revealed by histochemical staining) is responsible of root growth inhibition and not just to scavenge ROS.

2) I'm not sure if the conclusions derived from the cell type-specific activity of STOP1 (and consequently of ALMT1) are sufficiently justified. For instance, is the restored primary root sensitivity towards -P in pSCR::STOP1/stop1ko accompanied by increased Fe accumulation and peroxidase activity in the elongation zone?

In a new experiment, we have performed the iron staining in the complemented lines (P13 L391 and Fig. 6c and SupFig.19); this new result fully supports our initial conclusion of the restored primary root sensitivity towards -Pi (in addition to the reduced root growth and early differentiation of root hairs).

In addition, according to STOP1-promoter activity, this transcription factor is more strongly expressed in epidermal cells (Suppl Fig. 3a). Thus, why did the authors not try to restore STOP1 expression in this cell layer?

We did not find validated (i.e. published by independent laboratories) promoters active in all tissues of a specific zone along the proximo-distal axis of the root. We therefore used validated promoters specific for only one layer (i.e. endodermis, cortex). Anyway, not having tested the complementation in the epidermis does not invalidate the main conclusion that STOP1-dependent inhibition of root growth is in the distal part of the root tip and that STOP1 activity is not cell-specific and that malate probably diffuses in the root tip.

Moreover, the title of Fig.5 does not really make sense and should be related to the expression of STOP and not the cell types themselves.

We have changed the title of this figure (now Fig.6).

3) In general, the micrographs showing peroxidase activity in roots are of low quality. In most cases, only minor differences between peroxidase activity in roots of wild type and *almt1* and *stop1* mutants can be seen. As these results are of key relevance in the study, I would recommend improving the overall quality of these micrographs.

One alternative would be to enlarge the root pictures shown in Fig. 4j and k and/or take pictures of stained roots at higher magnifications. Perhaps some of the mutant alleles could be moved to a Suppl. Figure to increase space in Fig. 4. In addition, in order to establish whether the increased peroxidase activity precedes the inhibition of cell elongation, the enzymatic activity should also be assessed before cell elongation of Col-0 roots is severely arrested by P deficiency. In this regard, including an alternative approach to further demonstrate differences in peroxidase activity would also help to reinforce these datasets.

The .pdf files created are more pixelated than the original, high-quality files submitted. Anyway, we have increased the size of these micrographs (now Figure 5h). Assessing peroxidase activity with the 4-chloro-1-naphtol is quite delicate and we agree that only subtle differences are seen between seedlings with contrasted growth behaviours after two days in $-Pi$ (except *lpr1, lpr2* where the differential staining is high); a fortiori, we do not expect to observe convincing differential staining between seedlings grown only for a short time in $-Pi$. Also, we prefer to keep several alleles in the figure (Fig. 5h) to palliate this subtle difference between the mutants and the WT.

We have tried alternative staining like DAB and NBT, but in our hands they did not provide reproducible results.

4) The effects of malate, which plays a central role in the proposed model, have been poorly investigated leading to the following open questions: Are the changes in primary root length in response to exogenous malate supply statistically significant (Fig. 1d), and does external malate supply lead to a differential response of the EZ and RAM incl. corresponding Fe accumulation? Can the external supply of malate increase apoplastic Fe accumulation and peroxidase activity in the elongation zone of *almt1* and *stop1* mutants?

Statistics Fig 1d.: Done.

In a new experiment we have performed the staining of callose and iron in WT and the *almt1* mutant grown in $-Pi$ +malate. The result shows that exogenous malate complements the *almt1* mutant for callose and iron depositions in the elongation zone (Supplementary Fig. 9 and P9 L274), in addition to reduced growth. Thus, for three phenotypes examined, exogenous malate complements *almt1*.

Reviewer #4 (Remarks to the Author):

The authors have demonstrated the link between ALMT1 and STOP and Pi starvation. These two proteins have been already described then, the authors should pay more attention to the references included in the introduction regarding these two proteins (e.g.: STOP1 (<https://www.ncbi.nlm.nih.gov/pmc/articles/PMC2675709/>) and ALMT1 (<https://www.ncbi.nlm.nih.gov/pubmed/23624855/>)). A recent paper linking both proteins could be included (<https://www.ncbi.nlm.nih.gov/pubmed/25627216/>).

We added these missing references as well as other.

The figures are very complete but part of them are not used or described in the manuscript or sometime described very briefly late in the discussion. To illustrate this situation, the description of UPB1 appears in the last line of the discussion. To my opinion, UPB1 and the ROS homeostasis regulation could be included early in the paper and should be more discussed (<https://www.ncbi.nlm.nih.gov/pubmed/21074051/>).

We added a new result about ROS (P12 L373 and Supplementary Fig.11). We observed ROS in the transition and differentiation zone of WT in $-Pi$; these ROS staining appear less intense than in Müller et al (2015) (in particular in the meristem) probably because our protocol was less sensitive. This new experiment shows less accumulation of ROS in $+Pi+Fe$, $-Pi-Fe$ and $-Pi+Fe+SHAM$ conditions, compared to $-Pi+Fe$.

The results about UPB1 (P12 L365) are now presented in the Results part of the text.

The authors need to clarify the choice of the EMS mutants and justify the interest to present a large battery of *almt1* and *stop1* EMS mutants together with the two corresponding T-DNA mutant. The phenotypes are more or less similar then the use of one mutant is sufficient.

To strengthen the conclusions of our results (the histochemical staining of peroxidase activity is a good example, as discussed above) and to avoid working on phenotypes due to cryptic second-site mutations, we do our experiments as often as possible on several allelic mutants and on several independent transgenic lines. This is also a recommendation of a recent editorial in *The Plant Cell* journal (Buckler et al. "A Proposal Regarding Best Practices for Validating the Identity of Genetic Stocks and the Effects of Genetic Variants" *The Plant Cell* 28: 606–609, March 2016).

All the alleles we have used in this work were strong if not null alleles; the *almt1^{KO}* and *stop1^{KO}* lines were used to indicate the phenotype of these already published alleles and because they were the parental lines of some of our transgenic.

The phenotypic variation between the mutants is interesting and could be deepened regarding the expression of *almt1* and *stop1* in the different mutants. The lack of statistical analysis need to be corrected.

Statistical analysis: Done.

I am perhaps wrong, but the discussion ends with the idea of ROS homeostasis and peroxidases necessary for low phosphate root growth. Which is fine for me, but which is different from the initial message.

The conclusion is about the discovery of a new pathway (STOP1-ALMT1) and mechanism (based on exuded malate) inhibiting root growth in $-Pi$.

New and very interesting results are presented by the paper of Desnos and collaborators. But the quality and the readability of the paper will be largely improved with the reformatting of the figures and a better state of the art regarding ALMT1 and STOP1

We substantially modified the text, reformatted several figures (Fig.1, Fig.4, Fig.5, SupFig.2) and added several missing references.

Reviewer #5 (Remarks to the Author):

As per the editorial request, I am only commenting on the part of this paper in which the author's use AFM to characterize plant cell wall stiffening. I note 3 major concerns (points 3,5, 6 and 7 below).

I commend the authors for their method to secure the roots to the cover slip (and their clear description) since securely held samples is an important part of AFM methodology and is difficult to achieve.

1. However with living root samples it is also important to check that temporal changes did not occur purely due to the root drying out./ being affected by chemical interaction with the adhesive. Do the authors have data at e.g. 2 hrs in +Pi to indicate that stiffening was not occurring purely due to temporal changes in root morphology due to these sort of concerns? (e.g. +Pi data at 1 and 2h to compare with $-Pi$ data in Fig 4a)

The kinetic has been performed on agar plates, not on the microscope stage (P27 L850). Then, seedlings were mounted in a liquid medium on the microscope stage, therefore they did not dried out. It took around seven minutes to perform the ~ 100 measures of stiffness on each seedling.

We added a new figure with the stiffening kinetic in +Pi (Fig. 4a). This figure shows that transferring seedlings on a +Pi medium does not increase stiffness; instead it is the opposite. This indicates that the increasing stiffening observed in the kinetics in $-Pi$ is not due to the adhesive.

They state (line 698) that an average value of spring constant was 0.04 N/m.

2. To gain accurate force values it is necessary to use the precise spring constant of the AFM tip at the point of use. I think they may have done this since, from my reading of their reference 74, it is a part of the Yieldfinder software but it would be good to state this.

Indeed, spring constant of cantilevers were systematically evaluated before each series of measurements (by the method known as thermal tune). The value 0.04 N/m is a rounded average value observed among all our experiments. The nominal spring constant was 0.06 N/m. However, the proper identified value was systematically used during the recording of force-indentation curves.

Their method in which they adjusted the ramp size to give indentations of approximately the same depth for all samples is also to be commended since comparing indentations of similar size is an important step when comparing indentation data from samples of different stiffness.

3. A major question re the methodology in this paper is the location of targeted zone at ~ 500 μm from the “end root” (line 700) for all roots. If roots are a different size either through sample to sample variability of “identical roots”, or because root growth has been inhibited, is it correct that the transition zone in all the roots will be the same distance from the end of the root? The whole stiffening argument hangs on being sure that you are indenting like zone with like zone.

The AFM cantilever was positioned at ~500 μm from the end root, as measured with the crosshair. Since small variations can occur from root to root, we always positioned (using the optical camera) the cantilever just after the dense zone corresponding to the meristem. We therefore used two parameters to position the probe.

4. Lines 711-713 mentions that the fit was manually adjusted to optimise the fitting curve. Please could the authors provide some data to describe the goodness of fit they typically get. E.g. Ref 75 Fig 3 shows data points and a line of fit obtained. (I note in passing that this reference describes the formulation of the FORCE software for animal cells with measured stiffnesses in the range of 1-10kPa, one or two orders of magnitude compared to the stiffnesses being reported in this paper. The authors could consider commenting on this.

We have looked at all the fitted data and computed the average “goodness of fit” which is expressed as a root-mean-square deviation between the experimental and fitted curves. In average, we observed “goodness of fit” for all experiments about 0.47 ± 1.06 with a min value of 0.13 and a max value of 1.91 (median value of 0.24). The FORCE software was indeed developed for animal cells but this computer program only select areas on the force-indentation curve in order to compute a theoretical fit; thus, there is no connection between the software and the nature of the sample.

5. My second major point is that Line 224 claims that a higher stiffness is not significantly detected for *top1*, *almt1* and *lpr1:lpr2* lines. However the authors fail to comment that the *lpr1:lpr2* data presented actually indicates a significant softening of the cell wall in $-\text{Pi}$.

6. Line 225-226 makes the same “not higher stiffness” claim for WT grown in $-\text{P}$ and $-\text{Pi-Fe}$ but again fails to comment that that data present shows a significant softening.

The lower stiffness in WT grown in $-\text{Pi-Fe}$ compared to that in $-\text{Pi (+Fe)}$, is explained by our model whereby iron, through LPR ferroxidases and peroxidases activities, stimulates crosslinks in the cell wall.

However, it is true that we did not comment the reduced stiffness in *lpr1;lpr2* grown in $-\text{Pi}$ (Fig. 4b) and in the SHAM-treated WT (Fig. 5e) compared to $+\text{Pi}$ control; we therefore added a comment in the text (P12 L357). Our hypothesis is as follow: even in condition permissive for root growth, the cell wall contains crosslinks responsible of its basal strength; according to our results, this basal strength needs some activity of LPR ferroxidases and peroxidases. In the *lpr1;lpr2* double mutant, SHAM-treated WT or in WT seedlings grown in a Fe-deprived medium, the activities of LPR ferroxidases and/or peroxidases are strongly reduced and therefore less crosslinks are generated in the cell wall, thereby softening the cell wall. Accordingly, this softening is correlated with the strongly reduced peroxidase activity (Fig. 5g,h and Supplementary Fig. 10b).

7. My final major comment is that the authors, correctly, state that they are measuring transverse stiffness (line 216). However my understanding is that cell growth occurs mainly in the axial direction. In line 228 the paper suggests an simple causal scenario where $-\text{Pi}$ triggers the CW stiffening of pre-elongated cells to restrict their elongation but fails to comment on the large assumption that axial stiffness will be affected in the same way as transverse stiffness.

It's true that the word "transverse" was confusing (and it was also redundant with "root surface" in the same sentence) we therefore removed it.

Reviewers' comments:

Reviewer #1 (Remarks to the Author):

This is my second review of Balzergue et al, this time after resubmission to NCOMMS.

Its my opinion that the authors have done a great job in addressing my concerns and that of the other reviewers. Nice job! I think the paradox that malate exudation is important for continued root growth under low pH and aluminium and potentially high pH, whereas in -P conditions it inhibits growth, is fascinating.

I only have several minor comments and corrections.

Fig 1c – please can you better delineate the genotypes. It is confusing having so many marks on the x-axis, also mark significance?

Figure 3 – on the printed page I cannot see the deposition of Fe in Col -P condition that is different from the stop and almt mutants – however, this is fine on the screen

-Pi is first defined as phosphate limitation. Therefore all subsequent mentions of -Pi should be grammatically consistent. They are currently not, you may be able to get away with -Pi conditions not condition, you cannot say -Pi challenge, you should say under -Pi rather than in -Pi...etc. Check throughout.

L69 grammar - omit spatially and temporarily or put at the end of the sentence somehow

L81 – omit first sentence – this is really not required

L142 1b), we describe

L163 is consistent with rather than confirms

Supplementary Fig 3 is out of order

Fig 1 e – mark significance?

L346 – To my mind, this paragraph is written the wrong way round. What the authors are talking about is an induced artificial situation. I found it initially confusing being described this way. It may be easier to read if it talked about high peroxidase activity inhibiting growth under low Pi – this is what happens normally in the plant.

L381 – delete in it.

Paragraph starting L394 – this may suggest that ALMT1 is only properly active in these cells too?

L1372 -Pi prior to measuring

M. Gilliam

Reviewer #2 (Remarks to the Author):

The manuscript has been revised carefully. I have no criticism to this revision.

Reviewer #3 (Remarks to the Author):

In the revised version, the authors succeeded to sufficiently work out:

- a) ALMT1 expression is induced by P deficiency, in a STOP1-dependent manner. STOP1, in turn, is induced by -P at the post-transcriptional level.
- b) Supply of malate to *almt1* plants restores primary root sensitivity to -P, and restores Fe and callose accumulation in the elongation zone. Thus, the link between ALMT1-dependent malate extrusion under -P and inhibition of cell elongation appears strong.
- c) The differential Perls staining and callose deposition in *stop1* and *almt1* mutants, as compared to Col and *lrp1lpr2*, gives strong evidence that the deleterious effect of -P on the EZ and RAM is uncoupled in *stop1* and *almt1*. Therefore, malate appears to specifically target EZ.
- d) More inhibited cell elongation in the EZ under -P is shown to be associated with increased cell wall (CW) stiffening in Col-0. In *almt1* and *stop1* no significant changes in stiffening were detected.
- e) Based (purely) on a pharmacological approach, it is shown that P deficiency-induced CW stiffening and PR inhibition is dependent on the activity of peroxidases. Even though it still remains open how the peroxidase activity leads to cell wall stiffening, the present data justify the model drawn in Fig.7.

From a physiological perspective, it is still very hard to understand why malate should have such a specific role (e.g. compared to citrate) in Fe chelation and inhibition of cell elongation. As the -Pi-induced inhibition of cell elongation is highly reminiscent of the root response to Al, the question rises whether the agar medium used in all the experiments may have released Al under -Pi conditions. Given that P-deficient roots induce P-type ATPase and release more protons to the medium, it may well be that Al was set free from the agar. Actually, studies like the one in Gruber et al. 2013 (Plant Physiol.) have shown that some types of Sigma agar contain high concentrations of Al, which may have triggered the observed responses.

Reviewer #4 (Remarks to the Author):

The quality of the manuscript has been largely improved and the authors have answered to

the questions and comments made by the referees.
I still have one minor point which needs to be clarified

=> CM-H₂DCFDA is a probe used to detect the presence of H₂O₂ and peroxidases (the oxidation, by the enzyme in presence of H₂O₂, of fluorescein released a fluorescent oxidized product). The presence of SHAM reduce the fluorescence. ROS stands for several compounds, in this specific situation, the ROS detected is H₂O₂. This needs to be clarified.

And one comment without answer

The phenotypic variation between the mutants is interesting and could be deepened regarding the expression of *alm1* and *stop1* in the different mutants
=> No answer to this comment has been done

Reviewer #5 (Remarks to the Author):

I am very happy with the addition of a new figure showing the stiffening kinetic in +Pi which has been added. (I note that in the pdf title on same page as the figure there is a spelling mistake (Figure 4 : -Pi induces cell wall stiffening in the root transition zone) but this does not occur in the title at Line 1259).

The authors have also satisfied my concerns on the other points I raised.

Reviewer #1 (Remarks to the Author):

This is my second review of Balzergue et al, this time after resubmission to NCOMMS.

Its my opinion that the authors have done a great job in addressing my concerns and that of the other reviewers. Nice job! I think the paradox that malate exudation is important for continued root growth under low pH and aluminium and potentially high pH, whereas in -P conditions it inhibits growth, is fascinating.

I only have several minor comments and corrections.

Fig 1c – please can you better delineate the genotypes. It is confusing having so many marks on the x-axis, also mark significance?

We have enlarged the space between bars and added the significance.

Figure 3 – on the printed page I cannot see the deposition of Fe in Col -P condition that is different from the stop and almt mutants – however, this is fine on the screen

On the original file of Fig.3, the difference is clear (for submission, the file containing all figures as been compressed from 202 to 3Mo, this explains why some pictures appear less neat than in the original file).

-Pi is first defined as phosphate limitation. Therefore all subsequent mentions of -Pi should be grammatically consistent. They are currently not, you may be able to get away with -Pi conditions not condition, you cannot say -Pi challenge, you should say under - Pi rather than in -Pi...etc. Check throughout.

Done

L69 grammar - omit spatially and temporarily or put at the end of the sentence somehow

We removed them.

L81 – omit first sentence – this is really not required

Done

L142 1b), we describe

Corrected

L163 is consistent with rather than confirms

(now L163): corrected.

Supplementary Fig 3 is out of order

This Supplementary Figure has been moved at the end of the manuscript (now it is Sup Fig. 13). The corresponding text is now in the Discussion, L438.

Fig 1 e – mark significance?

Done.

L346 – To my mind, this paragraph is written the wrong way round. What the authors are talking about is an induced artificial situation. I found it initially confusing being described this way. It may be easier to read if it talked about high peroxidase activity inhibiting growth under low Pi – this is what happens normally in the plant.

(now L344): We replaced “decreased” by “high”, and “increased” by “reduced”.

L381 – delete in it.

(now L380): corrected.

Paragraph starting L394 – this may suggest that ALMT1 is only properly active in these cells too?

(now L393): Maybe. Alternatively, the non-complementing constructs are expressed too far away from the root tip to be able to complement the mutant.

L1372 –Pi prior to measuring

(now L1378): corrected.

M. Gilliam

Reviewer #2 (Remarks to the Author):

The manuscript has been revised carefully. I have no criticism to this revision.

Reviewer #3 (Remarks to the Author):

In the revised version, the authors succeeded to sufficiently work out:

- a) ALMT1 expression is induced by P deficiency, in a STOP1-dependent manner. STOP1, in turn, is induced by –P at the post-transcriptional level.
- b) Supply of malate to almt1 plants restores primary root sensitivity to –P, and restores Fe and callose accumulation in the elongation zone. Thus, the link between ALMT1-dependent malate extrusion under –P and inhibition of cell elongation appears strong.
- c) The differential Perl's staining and callose deposition in stop1 and almt1 mutants, as compared to Col and lrp1lpr2, gives strong evidence that the deleterious effect of –P on the EZ and RAM is uncoupled in stop1 and almt1. Therefore, malate appears to specifically target EZ.

d) More inhibited cell elongation in the EZ under -P is shown to be associated with increased cell wall (CW) stiffening in Col-0. In *almt1* and *stop1* no significant changes in stiffening were detected.

e) Based (purely) on a pharmacological approach, it is shown that P deficiency-induced CW stiffening and PR inhibition is dependent on the activity of peroxidases. Even though it still remains open how the peroxidase activity leads to cell wall stiffening, the present data justify the model drawn in Fig.7.

From a physiological perspective, it is still very hard to understand why malate should have such a specific role (e.g. compared to citrate) in Fe chelation and inhibition of cell elongation. As the -Pi-induced inhibition of cell elongation is highly reminiscent of the root response to Al, the question rises whether the agar medium used in all the experiments may have released Al under -Pi conditions. Given that P-deficient roots induce P-type ATPase and release more protons to the medium, it may well be that Al was set free from the agar. Actually, studies like the one in Gruber et al. 2013 (*Plant Physiol.*) have shown that some types of Sigma agar contain high concentrations of Al, which may have triggered the observed responses.

For information, below are indicated the conditions used by others to detect (in agar plates) the effect of Al³⁺ on root growth in Arabidopsis mutants hypersensitive to Al-toxicity:

- 1) Larsen et al. (*Plant & Soil* 192:3, 1997) : agar plate soaked with 250 to 1500 μM AlCl₃ (pH4.2);
- 2) Gabrielson et al. (*J.Exp.Bot.* 57:943, 2006) : plates soaked with 750 μM AlCl₃ (pH 4.2).
- 3) Hoekenga et al. (*PNAS* 103: 9738, 2006): 500 μM AlCl₃ (pH 4.2) ;
- 4) Zhao et al. (*PLoS One* 6: e28086, 2011) : 50 to 300 μM AlCl₃ (pH4-5.6);

As we can see, in agar plates (where Al³⁺ is thought to be trapped in agar) we need high concentrations of Al³⁺ and a very low pH (necessary for Al³⁺ solubility). Note that in these conditions the WT still exhibits a long root compared to the WT grown in our -Pi condition.

In the agar we used in our experiments (8g/L), the total aluminum concentration is 13,2 μM (as measured by two independent ICP analyses). Note that we don't know in which chemical form is this aluminum: is it soluble (Al³⁺), or is it in a biologically more inert, insoluble form (aluminum silicates, aluminum oxides, etc.)? [the article of Gruber et al. (*Plant Physiol.* 163:161, 2013) does not discuss this point].

It is therefore clear that our -Pi agar plates are far from the conditions necessary to inhibit root growth by Al³⁺.

Furthermore, even by assuming that all this Al is soluble (which is not known), we strongly don't think it could cause the inhibition of WT root growth in -Pi for the following reasons:

- 1) If this concentration were sufficient to alter root growth in WT seedlings, *a fortiori* we would have seen a strong reduction of the primary root length of the *stop1* and *almt1* mutants because these mutants are extremely sensitive to Al³⁺ toxicity (*PNAS* 104:9900, 2007; *Plant Physiol.*, 150:281, 2009). This is not the case in our -Pi plates; instead we

observe the opposite: the *stop1* and *almt1* mutants have a primary root longer than that of the WT.

2) We observe the -Pi-dependent root growth inhibition when WT seedlings are grown in liquid medium (therefore without Al³⁺).

Reviewer #4 (Remarks to the Author):

The quality of the manuscript has been largely improved and the authors have answered to the questions and comments made by the referees.

I still have one minor point which needs to be clarify

=> CM-H2DCFDA is a probe used to detect the presence of H2O2 and peroxidases (the oxidation, by the enzyme in presence of H2O2, of fluorescein released a fluorescent oxidized product). The presence of SHAM reduce the fluorescence . ROS stands for several compounds, in this specific situation, the ROS detected is H2O2. This needs to be clarified.

L375: In the main text we added: "(H₂O₂ and peroxidases-mediated oxidation of CM-H2DCFDA)".

L846: In the Method part we added "this compound is oxidized by peroxidases in presence of H₂O₂.

And one comment without answer

The phenotypic variation between the mutants is interesting and could be deepened regarding the expression of *almt1* and *stop1* in the different mutants

=> No answer to this comment has been done

Since all the alleles described in this manuscript are strong (if not null), we think that the phenotypic variability in the primary root length is due to parasite mutations.

Reviewer #5 (Remarks to the Author):

I am very happy with the addition of a new figure showing the stiffening kinetic in + Pi which has been added. (I note that in the pdf title on same page as the figure there is a spelling mistake (Figure 4 : -Pi induces cell wall stiffnening in the root transition zone) but this does not occur in the title at Line 1259).

Title of Figure 4: corrected.

The authors have also satisfied my concerns on the other points I raised.